# Learning Graph Structure With A Finite-State Automaton Layer

**Daniel D. Johnson, Hugo Larochelle, Daniel Tarlow**
Google Research
{ddjohnson, hugolarochelle, dtarlow}@google.com

## Abstract

Graph-based neural network models are producing strong results in a number of
domains, in part because graphs provide flexibility to encode domain knowledge
in the form of relational structure (edges) between nodes in the graph. In practice,
edges are used both to represent intrinsic structure (e.g., abstract syntax trees of
programs) and more abstract relations that aid reasoning for a downstream task
(e.g., results of relevant program analyses). In this work, we study the problem of
learning to derive abstract relations from the intrinsic graph structure. Motivated by
their power in program analyses, we consider relations defined by paths on the base
graph accepted by a finite-state automaton. We show how to learn these relations
end-to-end by relaxing the problem into learning finite-state automata policies on a
graph-based POMDP and then training these policies using implicit differentiation.
The result is a differentiable Graph Finite-State Automaton (GFSA) layer that adds
a new edge type (expressed as a weighted adjacency matrix) to a base graph. We
demonstrate that this layer can find shortcuts in grid-world graphs and reproduce
simple static analyses on Python programs. Additionally, we combine the GFSA
layer with a larger graph-based model trained end-to-end on the variable misuse
program understanding task, and find that using the GFSA layer leads to better
performance than using hand-engineered semantic edges or other baseline methods
for adding learned edge types.

## 1   Introduction

Determining exactly which relationships to include when representing an object as a graph is not
always straightforward. As a motivating example, consider a dataset of source code samples. One
natural way to represent these as graphs is to use the abstract syntax tree (AST), a parsed version of
the code where each node represents a logical component.[1] But one can also add additional edges to
each graph in order to better capture program behaviors. Indeed, adding additional edges to represent
control flow or data dependence has been shown to improve performance on code-understanding
tasks when compared to a AST-only or token-sequence representation [1, 19].

An interesting observation is that these additional edges are fully determined by the AST, generally
by using hand-coded static analysis algorithms. This kind of program abstraction is reminiscent
of temporal abstraction in reinforcement learning (e.g., action repeats or options [30, 38]). In both
cases, derived higher-level relationships allow reasoning more abstractly and over longer distances
(in program locations or time).

In this work, we construct a differentiable neural network layer by combining two ideas: program
analyses expressed as reachability problems on graphs [34], and mathematical tools for analyzing
temporal behaviors of reinforcement learning policies [13]. This layer, which we call a Graph

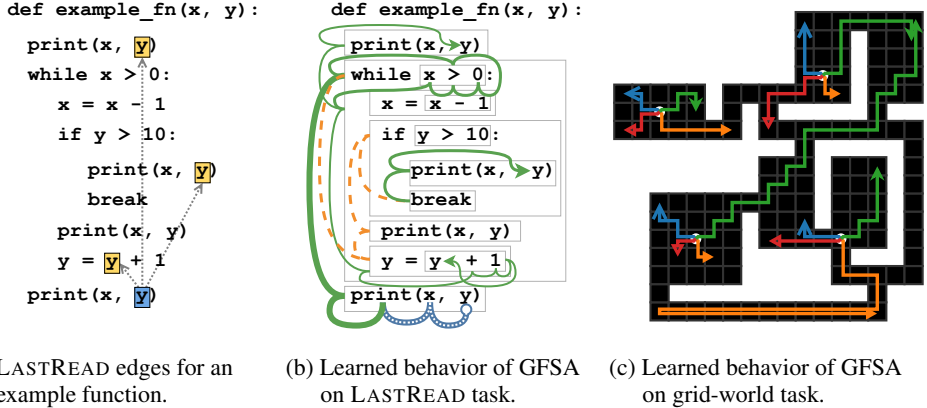

(a) LASTREAD edges for an example function.

(b) Learned behavior of GFSA on LASTREAD task.

(c) Learned behavior of GFSA on grid-world task.

Figure 1: (a) Target edges for the LASTREAD task starting from the final use of y on a handwritten example function. (b) Learned behavior starting at the final use of y (blue circle). Thickness represents probability mass, color and style represent the finite-state memory, and boxes represent AST nodes in the graph. The automaton changes to a reverse-execution mode (green) and steps backward to the while loop, then nondeterministically either looks at the condition or switches to a break-finding mode (orange) and jumps to the body. In the first case, it checks for uses of y in the condition, then splits again between the previous print and the loop body. In the second, it walks upward until finding a break statement, then transitions back to the reverse-execution mode. For simplicity, we hide backtracking trajectories and combine some intermediate steps. Note that only the start and end locations (colored boxes in (a)) are supervised; all intermediate steps are learned. (c) Colored arrows denote the path taken by the GFSA policy for each option, shown starting from four arbitrary start positions (white) on a grid-world layout not seen during training. The tabular agent can jump from each start position to the endpoint of any of its arrows in a single step.

Finite-State Automaton (GFSA), can be trained end-to-end to add derived relationships (edges) to arbitrary graph-structured data based on performance on a downstream task.[2] We show empirically that the GFSA layer has favorable inductive biases relative to baseline methods for learning edge structures in graph-based neural networks.

## 2 Background

### 2.1 Neural Networks on Graphs

Many neural architectures have been proposed for graph-structured data. We focus on two general families of models: first, message-passing neural networks (MPNNs), which compute sums of messages sent across each edge [17], including recurrent models such as Gated Graph Neural Networks (GGNNs) [27]; second, transformer-like models operating on the nodes of a graph, which include the Relation-Aware Transformer (RAT) [42] and Graph Relational Embedding Attention Transformer (GREAT) [19] models along with other generalizations of relative attention [37]. All of these models assume that each node is associated with a feature vector, and each edge is associated with a feature vector or a discrete type.

### 2.2 Derived Relationships as Constrained Reachability

Compilers and static analysis tools use a variety of techniques to analyze programs, many of which are based on fixed-point analysis over a problem-specific abstract lattice (see for instance Cousot and Cousot [10]). However, it is possible to recast many of these analyses within a different framework: graph reachability under formal language constraints [34].

Consider a directed graph $G$ where the nodes and edges are annotated with labels from finite sets $\mathcal{N}$ and $\mathcal{E}$. Let $L$ be a formal language over alphabet $\Sigma = \mathcal{N} \cup \mathcal{E}$, i.e. $L \subseteq \Sigma^*$ is a set of words (finite

sequences of labels from $\Sigma$) built using formal rules. One useful family of languages is the set of *regular languages*: a regular language $L$ consists of the words that match a regular expression, or equivalently the words that a finite-state automaton (FSA) accepts [20]. Note that each path in $G$ corresponds to a word in $\Sigma^*$, obtained by concatenating the node and edge labels along the path. We say that a path from node $n_1$ to $n_2$ is an $L$-path if this word is in $L$.

Using a construction similar to Reps [34], one can construct a regular language $L$ such that, if $n_1$ is the location of a variable and $n_2$ is the location of a previous read from that variable, then there is an $L$-path from $n_1$ to $n_2$ (and every $L$-path is of this form); roughly, $L$ contains paths that trace backward through the program's execution. This corresponds to edge type LASTREAD as described by Allamanis et al. [1], which is visualized in Figure 1a. Similarly, one can define edges and corresponding regular languages that connect each use of a variable to the possible locations of the previous assignment to it (the LASTWRITE edge type from [1]) or connect each statement to the statements that could execute directly afterward (which we denote NEXTCONTROLFLOW since it corresponds to the control flow graph); see appendix A.1 for details. More generally, the existence of $L$-paths summarizes the presence of longer chains of relationships with a single pairwise relation. Depending on $L$, this can represent transitive closures, compositions of edges, or other more complex patterns. We may not always know which language $L$ would be useful for a task of interest; our approach makes it possible to jointly learn $L$ and use the $L$-paths as an abstraction for the downstream task.

# 3 Approach

Consider, as a motivating example, a graph representing the abstract syntax tree for a Python program. Each of the nodes of this tree has a type, for instance "identifier", "binary operation" or "if statement", and edges correspond to AST fields such as "X is the left child of Y" or "X is the loop condition of Y". Section 2.2 suggests that $L$-paths on this graph are a useful abstraction for higher-level reasoning, but we do not know what the best choice of $L$ is; we seek a mechanism to learn it end-to-end.

We propose placing an agent on a node of this tree, with actions corresponding to the possible fields of each node (e.g. "go to parent node" or "go to left child"), and observations giving local information about each node (e.g. "this is an identifier, but not the one we are trying to analyze"). Note that the trajectories of this agent then correspond to paths in the graph. We allow the agent to terminate the episode and add an edge from its initial to its current location, thus "accepting" the path it has taken. By averaging over all trajectories, we obtain an expected adjacency matrix for these edges that summarizes the paths that the agent tends to accept, which we use as an output edge type.

If the agent's actions were determined by a finite-state automaton for a regular language $L$, the added edges would correspond to $L$-paths. We propose parameterizing the agent with a *learnable* finite-state automaton, so that it can learn to do the kinds of analyses that a regular language can express. As long as the actions and observations are shared across all ASTs, we can then apply this policy to many different ASTs, even ones not seen at training time.

In this section, we formalize and generalize this intuition by describing a transformation from graphs into partially-observable Markov decision processes (POMDPs). We show that, for agents with a finite-state hidden memory, we can efficiently compute and differentiate through the distribution of trajectory endpoints. We propose using this distribution to define a new edge type, and demonstrate that any regular-language-constrained reachability problem (and in particular, basic program analyses) can be expressed as a policy of this form.

## 3.1 From Graphs to POMDPs

Suppose we have a family of graphs $\mathcal{G}$ with an associated set of node types $\mathcal{N}$. Our approach is to transform each graph $G \in \mathcal{G}$ to a rewardless POMDP, in which an agent takes a sequence of actions to move between nodes of the graph while observing only local information about its current location. To ensure that all graphs have compatible action and observation spaces, for each node type $\tau(n) \in \mathcal{N}$ we choose a finite set $\mathcal{M}_{\tau(n)}$ of movement actions associated with that node type (e.g. the set of possible fields that can be followed) and a finite set $\Omega_{\tau(n)}$ of observations (which include the node type as well as other task-specific information). These choices may depend on domain knowledge about the graph family or the task to be solved; see appendix B for the specific choices we used in our experiments.

At each node $n_t \in N$ of a graph $G \in \mathcal{G}$, the agent selects an action $a_t$ from the set

$$\mathcal{A}_t = \big\{ (\text{MOVE}, m) \mid m \in \mathcal{M}_{\tau(n_t)} \big\} \sqcup \{ \text{ADDEDGEANDSTOP}, \text{STOP}, \text{BACKTRACK} \}$$

according to some policy $\pi$. If $a_t = \text{ADDEDGEANDSTOP}$, the episode terminates by adding an edge $(n_0, n_t)$ to the output adjacency matrix. If $a_t = \text{STOP}$, the episode terminates without adding an edge. If $a_t = (\text{MOVE}, m)$, the agent is either moved to an adjacent node $n_{t+1} \in N$ in the graph by the environment or stays at node $n_t$ (and thus $n_{t+1} = n_t$), and then receives an observation $\omega_{t+1}$ with $\omega_{t+1} \in \Omega_{\tau(n_{t+1})}$. The MDP is partially observable because the agent does not see node identities or the global structure; instead, $\omega_{t+1}$ encodes only node types and other local information. Since derived edge types may depend on existing pairwise relationships between nodes (for instance, whether two variables have the same name), we allow the observations to depend on the initial node $n_0$ as well as the current node $n_t$ and most recent transition; in effect, each choice of $n_0 \in N$ specifies a different version of the POMDP for graph $G$. Finally, if $a_t = \text{BACKTRACK}$ is selected, the agent is reset to its initial state. We note that, since the action and observation spaces are shared between all graphs in $\mathcal{G}$, a single policy $\pi$ can be applied to any graph $G \in \mathcal{G}$.

We would like our agent to be powerful enough to extract useful information from this POMDP, but simple enough that we can efficiently compute and differentiate through the learned trajectories. Since our motivating program analyses can be represented as regular languages, which correspond to finite-state automata (see section 2.2), we focus on agents augmented with a finite-state memory.

### 3.2 Computing Absorbing Probabilities

Here we describe an efficient way to compute and differentiate through the distribution over trajectory endpoints for a finite-state memory policy over a graph. Let $Z$ be a finite set of memory states, and consider a specific policy $\pi_\theta(a_t, z_{t+1} \mid \omega_t, z_t)$ parameterized by $\theta$ (see appendix C.1 for details regarding the parameterization we use). Combining the policy $\pi_\theta$ with the environment dynamics for a single graph $G$ yields an absorbing Markov chain over tuples $(n_t, \omega_t, z_t)$, with transition distribution

$$p(n_{t+1}, \omega_{t+1}, z_{t+1} | n_t, \omega_t, z_t, n_0) = \sum_{m_t} \pi_\theta \big( a_t = (\text{MOVE}, m_t), z_{t+1} | \omega_t, z_t \big)$$
$$\cdot p(n_{t+1} | n_t, m_t) \cdot p(\omega_{t+1} | n_{t+1}, n_t, m_t, n_0)$$

and halting distribution $\pi_\theta(a_t \in \{\text{ADDEDGEANDSTOP}, \text{STOP}, \text{BACKTRACK}\} \mid n_t, \omega_t, z_t)$. We can represent this distribution via a transition matrix $Q_{n_0} \in \mathbb{R}^{K \times K}$ where $K$ is the set of possible $(n, \omega, z)$ tuples, along with a halting matrix $H \in \mathbb{R}^{(3 \times |N|) \times K}$ (keeping track of the final node $n_T \in N$ as well as the halting action). We can then compute probabilities for each final action by summing over each possible trajectory length $i$:

$$p(a_T, n_T | n_0, \pi_\theta) = \left[ \sum_{i \geq 0} H Q_{n_0}^i \boldsymbol{\delta}_{n_0} \right]_{(a_T, n_T)} = H_{(a_T, n_T),:} \, (I - Q_{n_0})^{-1} \boldsymbol{\delta}_{n_0}, \qquad (1)$$

where $\boldsymbol{\delta}_{n_0}$ is a vector with a 1 at the position of the initial state tuple $(n_0, \omega_0, z_0)$. Note that, since the matrix depends on the initial state $n_0$, it would be inefficient to analytically invert this matrix for every $n_0$. We thus use $T_{\max}$ iterations (typically 128) of the the Richardson iterative solver [3] to obtain an approximate solution using only efficient matrix-vector products; this is equivalent to truncating the sum to include only paths of length at most $T_{\max}$.

To compute gradients with respect to $\theta$, we use implicit differentiation to express the gradients as the solution to another (transposed) linear system and use the same iterative solver; this ensures that the memory cost of this procedure is independent of $T_{\max}$ (roughly the cost of a single propagation step for a message-passing model). We implement the forward and backward passes using the automatic differentiation package JAX [8], which makes it straightforward to use implicit differentiation with an efficient matrix-vector product implementation that avoids materializing the full transition matrix $Q_{n_0}$ for each value of $n_0$ (see appendix C for details).

### 3.3 Absorbing Probabilities as a Derived Adjacency Matrix

Finally, we construct an output weighted adjacency matrix by averaging over trajectories:

$$\widehat{A}_{n,n'} = p(a_T = \text{ADDEDGEANDSTOP}, n_T = n' \mid n_0 = n, a_T \neq \text{BACKTRACK}, z_0, \pi_\theta),$$
$$A_{n,n'} = \sigma\big( a\, \sigma^{-1}\big(\widehat{A}_{n,n'}\big) + b \big) \qquad (2)$$

where $a, b \in \mathbb{R}$ are optional learned adjustment parameters, $\sigma$ denotes the logistic sigmoid, and $\sigma^{-1}$ denotes its inverse. Note that, since they are derived from a probability distribution, the columns of $\widehat{A}_{n,n'}$ sum to at most 1. The adjustment parameters $a$ and $b$ remove this restriction, allowing the model to express one-to-many relationships.

Given a fixed initial automaton state $z_0$, $A_{n,n'}$ can be viewed as a new weighted edge type. Since $A_{n,n'}$ is differentiable with respect to the policy parameters $\theta$, this adjacency matrix can either be supervised directly or passed to a downstream graph model and trained end-to-end.

### 3.4 Connections to Constrained Reachability Problems

As described in section 2.2, many interesting derived edge types can be expressed as the solutions to constrained reachability problems. Here, we describe a correspondence between constrained reachability problems on graphs and trajectories within the POMDPs defined in section 3.1.

**Proposition 1.** *Let $\mathcal{G}$ be a family of graphs annotated with node and edge types. There exists an encoding of graphs $G \in \mathcal{G}$ into POMDPs as described in section 3.1 and a mapping from regular languages $L$ into finite-state policies $\pi_L$ such that, for any $G \in \mathcal{G}$, there is an $L$-path from $n_0$ to $n_T$ in $G$ if and only if $p(a_T = \text{ADDEDGEANDSTOP}, n_T | n_0, \pi_L) > 0$.*

In other words, for any ordered pair of nodes $(n_0, n_T)$, determining if there is a path in $G$ that satisfies regular-language reachability constraints is equivalent to determining if a specific policy takes the ADDEDGEANDSTOP action at node $n_T$ with nonzero probability when started at node $n_0$, under a particular POMDP representation. See appendix A.2 for a proof. As a specific consequence:

**Corollary.** *There exists an encoding of program AST graphs into POMDPs and a specific policy $\pi_{\text{NEXT-CF}}$ with finite-state memory such that $p(a_T = \text{ADDEDGEANDSTOP}, n_T \mid n_0, \pi) > 0$ if and only if $(n_0, n_T)$ is an edge of type NEXTCONTROLFLOW in the augmented AST graph. Similarly, there are policies $\pi_{\text{LAST-READ}}$ and $\pi_{\text{LAST-WRITE}}$ for edges of type LASTREAD and LASTWRITE, respectively.*

### 3.5 Connections to Reinforcement Learning and the Successor Representation

The GFSA layer deterministically computes continuous edge weights by marginalizing over trajectories. These weights can then be transformed nonlinearly (e.g. $f(\mathbb{E}[\tau])$ where $f$ is the downstream model and loss and $\tau$ are edge additions from trajectories). In contrast, standard RL approaches produce stochastic discrete samples. As such, is not possible to "drop in" an RL approach instead of GFSA; one must first reformulate the model and task in terms of an expected reward $\mathbb{E}[f(\tau)]$.

Even so, there are interesting connections between the gradient updates for GFSA and traditional RL. In particular, the columns of the matrix $(I - Q_{n_0})^{-1}$ are known in the RL literature as the *successor representation*. If immediate rewards are described by $r$, then taking a product $r^T (I - Q_{n_0})^{-1}$ corresponds to computing the value function [13]. In our case, instead of specifying a reward, we use the GFSA layer for a downstream task that requires optimizing some loss $\mathcal{L}$. When computing gradients of our parameters with respect to $\mathcal{L}$, backpropagation computes a linear approximation of the downstream network and loss function and then uses it in the intermediate expression

$$\frac{\partial \mathcal{L}}{\partial p(\cdot | n_0, \pi_\theta)}^T H (I - Q_{n_0})^{-1}.$$

This is analogous to a non-stationary "reward function" for the GFSA policy, which assigns reward to the absorbing states that produce useful edges for the rest of the model. Unlike in standard RL, however, this quantity depends on the full marginal distribution over behaviors. As such, the "reward" assigned to a given trajectory may depend on the probability of other, mutually exclusive trajectories.

## 4 Related Work

Some prior work has explored learning edges in graphs. Kipf et al. [25] propose a neural relational inference model, which infers pairwise relationships from observed particle trajectories but does not add them to a base graph. Franceschi et al. [15] infer missing edges in a single fixed graph by jointly optimizing the edge structure and a classification model; this method only infers edges of a predefined type, and does not generalize to new graphs at test time. Yun et al. [48] propose adding

new edge types to a graph family by learning to compose a fixed number of existing edge types, which can be seen as a special case of GFSA where each state is visited once. The MINERVA model, described in Das et al. [12], uses an RL agent trained with REINFORCE to add edges to a knowledge base, but requires direct supervision of edges. Wang et al. [43] use a RL policy to *remove* existing edges from a noisy graph, with reward coming from a downstream classification task.

Bielik et al. [7] apply decision trees to program traces with a counterexample-guided program generator in order to learn static analyses of programs. Their method is provably correct, but cannot be used as a component of an end-to-end differentiable model or applied to general graph structures.

Our work shares many commonalities with reinforcement learning techniques. Section 3.5 describes a connection between the GFSA computation and the successor representation [13]. Our work is also conceptually similar to methods for learning options. For instance, Bacon et al. [4] describe an end-to-end architecture for learning options by differentiating through a primary policy's reward. Their option policies and primary policy are analogous to our GFSA edge types and downstream model; on the other hand, they apply policy gradient methods to trajectory samples instead of optimizing over full marginal distributions, and their full architecture is still a policy, not a general model on graphs.

Existing graph embedding methods have used stochastic walks on graphs [47, 23, 49, 24, 21, 9], but generally assume uniform random walks. Alon et al. [2] propose representing ASTs by sampling random paths and concatenating their node labels, then attending over the resulting sequences. Dai et al. [11] describe a framework of MDPs over graphs, but focus on a "learning to explore" task, where the goal is to visit many nodes and the agent can see the entire subgraph it has already visited. Hudson and Manning [22] propose treating the nodes of an inferred scene graph as states of a learned state machine, and learning to update the current active node based on natural-language inputs.

Self-attention can be viewed as constructing a weighted adjacency matrix similar to GFSA, but only considers pairwise relationships and not longer paths. Existing approaches to learning multi-step path-based relationships include iterating a graph neural network until convergence [36] and using a learned stopping criterion as in Universal Transformers [14]. The algorithm in section 3.2 in particular resembles running a separate graph neural network model to convergence for each start node and training with recurrent backpropagation [36, 28], and is also similar to other uses of implicit differentiation [45, 5, 32]. The GFSA layer enables multi-step relationships to be efficiently computed for every start node in parallel and provides good expressivity and inductive biases for learning edges, in contrast to previous techniques that focus on learning node representations and must learn from scratch to propagate multi-step information without letting distinct paths interfere with each other.

Weiss et al. [44] describe a method for extracting a discrete finite-state automaton from a RNN; this assumes access to an existing trained RNN for the task, and is intended for recognizing sequences, not adding edges to graphs. See also Mohri [31] for a framework of weighted automata on sequences.

## 5 Experiments

### 5.1 Grid-World Options

As an illustrative example, we consider the task of discovering useful navigation strategies in grid-world environments. We generate grid-world layouts using the LabMaze generator [6],[3] and interpret each cell as a node in a graph $G$, where edges represent cardinal directions. We augment this graph with additional edges from a GFSA layer, using four independent GFSA policies to add four additional edge types; let $G'_\theta(G)$ denote the augmented graph using GFSA parameters $\theta$. Next, we construct a pathfinding task on the augmented graph $G'_\theta(G)$, in which a graph-specific agent finds the shortest path to some goal node $g$. We assign an equal cost to all edges (including those that the GFSA layer adds); when the agent follows a GFSA edge, it ends up at a destination cell with probability proportional to the edge weights from the GFSA layer.

Inspired by existing work on meta-learning options [16], we interpret the GFSA-derived edges as a kind of option for this agent: given a random graph, the edges added by the GFSA layer should make it possible to quickly reach any goal node $g$ from any start location $n_0$. More specifically, we train the graph-independent GFSA layer (in an outer loop) to minimize the number of steps that a

graph-specific policy (trained in an inner loop) takes to reach the goal $g$, i.e. we minimize

$$\mathcal{L} = \mathbb{E}_{G,n_0,g} \left[ \mathbb{E}_{n_t \sim \pi^*(\cdot | n_{t-1}, g, G'_\theta(G))} \left[ T \mid n_T = g \right] \right]$$

where $\pi^*(\cdot | g, G'_\theta(G))$ is an optimal tabular policy for graph $G'_\theta(G)$ and goal $g$. In order to differentiate this with respect to the GFSA parameters $\theta$, we use entropy regularization to ensure $\pi^*(\cdot | g, G'_\theta(G))$ is smooth, and solve for it by iterating the soft Bellman equation until convergence [18], again using implicit differentiation to backpropagate through that solution (see appendix D.1).

Figure 1c shows the derived edges learned by the GFSA layer on a graph not seen during training; we find that the edges learned by the GFSA layer are discrete and roughly correspond to diagonal motions in the grid. Over the course of training the GFSA layer, the average number of steps taken by the (optimal) primary policy (on a validation set of unseen layouts) decreases from 40.1 steps to 11.5 steps, a substantial improvement in the end-to-end performance. This example illustrates the kind of relationships the GFSA layer can learn from end-to-end supervision; note that we do not claim these options are optimal for this task or would be practical in a more traditional RL context.

## 5.2   Learning Static Analyses of Python Code

Proposition 1 ensures that a GFSA is theoretically capable of performing simple static analyses of code. We demonstrate that the GFSA can practically learn to do these analyses by casting them as pairwise binary classification problems. We first generate a synthetic dataset of Python programs by sampling from a probabilistic context-free grammar over a subset of Python. We then transform the corresponding ASTs into graphs, and compute the three edge types NEXTCONTROLFLOW, LASTREAD, and LASTWRITE, which are commonly used for program understanding tasks [1, 19] and which we describe in section 2.2. Note that there may be multiple edges from the same statement or variable, since there are often multiple possible execution paths through the program.

For each of these edge types, we train a GFSA layer to classify whether each ordered pair of nodes is connected with an edge of that type. We use the focal-loss objective [29], a more stable variant of the cross-entropy loss for highly unbalanced classification problems, minimizing

$$\mathcal{L} = \mathbb{E}_{(N,E) \sim \mathcal{D}} \left[ \sum_{n_1,n_2 \in N} \begin{cases} -(1 - A_{n_1,n_2})^\gamma \log(A_{n_1,n_2}) & \text{if } (n_1 \to n_2) \in E, \\ -(A_{n_1,n_2})^\gamma \log(1 - A_{n_1,n_2}) & \text{otherwise} \end{cases} \right]$$

where the expectation is taken over graphs in the training dataset $\mathcal{D}$.

We compare against four graph model baselines: a GGNN [27], a GREAT model over AST graphs [19], a RAT model [42], and an NRI-style encoder [25]. For the GGNN, GREAT, and RAT models, we present results for two methods of computing output adjacency matrices: the first computes a learned key-value dot product (similar to dot-product attention) and interprets it as an adjacency matrix, and the second runs the model separately for each possible source node, tagging that source with an extra node feature, and computing an output for each possible destination (denoted "nodewise"). For the NRI encoder model, the output head is an MLP over node feature pairs as described by Kipf et al. [25]; we extend the NRI model with residual connections and layer normalization to improve stability, similar to a transformer model [40]. All baselines use a logistic sigmoid as a final activation, and are trained with the focal-loss objective. See appendix D.2 for more details.

As an ablation, we also train a standard RL agent with the same parameterization as GFSA, inspired by MINERVA [12]. We replace the cross-entropy loss with a reward of +1 for adding a correct edge (or correctly not adding any) and 0 otherwise, and train using REINFORCE with 20 rollouts per start node and a leave-one-out control variate [46, 26]. Since edges are added by single trajectories rather than marginals over trajectories, this RL agent can add at most one edge from each start node.

Table 1 shows results of each of these models on the three edge classification tasks. We present results after training on a dataset of 100,000 examples as well as on a smaller dataset of only 100 examples, and report F1 scores at the best classification threshold; we choose the model with the best validation performance from a 32-job random hyperparameter search. To assess generalization, we also show results on two modified data distributions: programs of half the size of those in the training set (0.5x), and programs twice the size (2x). When trained on 100,000 examples, all models achieve high accuracy on examples of the training size, but some fail to generalize, especially to larger programs. When trained on 100 examples, only the GFSA layer and RL ablation consistently achieve

Table 1: Results on the program analysis edge-classification tasks. Values are F1 scores (in percent), with bold indicating overlapping 95% confidence intervals with the best model; see appendix D.2.3 for full-precision results. "nw" denotes nodewise output, and "dp" denotes dot-product output.

| | **100,000 training examples** | | | | | | | | |
|---|---|---|---|---|---|---|---|---|---|
| **Task** | Next Control Flow | | | Last Read | | | Last Write | | |
| **Example size** | 1x | 2x | 0.5x | 1x | 2x | 0.5x | 1x | 2x | 0.5x |
| *RAT nw* | 99.98 | 99.94 | 99.99 | 99.86 | 96.29 | **99.98** | 99.83 | 94.87 | **99.97** |
| *GREAT nw* | 99.98 | 99.87 | 99.98 | 99.91 | 95.12 | **99.98** | 99.75 | 93.22 | 99.93 |
| *GGNN nw* | 99.98 | 93.90 | 97.77 | 95.52 | 9.22 | 86.24 | 98.82 | 40.69 | 88.28 |
| *RAT dp* | 99.99 | 92.53 | 96.59 | 99.96 | 42.58 | 91.96 | 99.98 | 68.96 | 99.76 |
| *GREAT dp* | 99.99 | 96.32 | 98.36 | **99.99** | 47.07 | 99.78 | **99.99** | 68.46 | 99.88 |
| *GGNN dp* | 99.94 | 62.75 | 98.51 | 98.44 | 0.99 | 63.77 | 99.35 | 38.40 | 94.52 |
| *NRI encoder* | 99.98 | 85.91 | 99.92 | 99.83 | 43.44 | 99.39 | 99.87 | 52.73 | 99.84 |
| *RL ablation* | 94.24 | 93.56 | 94.83 | 96.69 | 94.85 | 97.85 | 98.08 | 96.64 | 98.93 |
| *GFSA (ours)* | **100.00** | **99.99** | **100.00** | 99.66 | **98.94** | 99.90 | 99.47 | **98.73** | 99.78 |
| | **100 training examples** | | | | | | | | |
| *RAT nw* | 98.63 | 95.93 | 96.32 | 80.28 | 1.12 | 83.49 | 79.27 | 8.91 | 83.79 |
| *GREAT nw* | 98.23 | 97.98 | 98.52 | 78.88 | 6.96 | 60.90 | 80.19 | 40.22 | 84.54 |
| *GGNN nw* | 99.37 | 98.36 | 98.60 | 79.36 | 28.28 | 5.66 | 91.13 | 71.62 | 91.79 |
| *RAT dp* | 81.81 | 68.46 | 87.05 | 59.53 | 28.91 | 62.27 | 75.99 | 48.10 | 81.63 |
| *GREAT dp* | 86.60 | 62.98 | 80.58 | 57.02 | 27.13 | 64.48 | 73.69 | 46.27 | 80.03 |
| *GGNN dp* | 76.85 | 22.99 | 28.91 | 44.37 | 9.64 | 38.34 | 53.82 | 17.84 | 55.08 |
| *NRI encoder* | 81.74 | 69.08 | 88.87 | 68.69 | 26.64 | 73.52 | 65.38 | 36.43 | 73.86 |
| *RL ablation* | 91.70 | 91.14 | 92.29 | 98.48 | 97.03 | 99.17 | 98.32 | **96.96** | 99.07 |
| *GFSA (ours)* | **99.99** | **99.99** | **100.00** | **98.81** | **97.82** | **99.22** | **98.71** | **96.98** | **99.55** |

high accuracy, highlighting the strong inductive bias for constrained-reachability-based reasoning tasks. The GFSA layer trained with exact marginals and cross-entropy loss obtains higher accuracy than the RL ablation, and also converges more reliably: 82% of GFSA layer training jobs achieve at least 90% accuracy on the validation set, compared to only 11% of RL ablation jobs.

Figure 1b shows an example of the behavior that the GFSA layer learns for the LASTREAD task based on only input-output supervision. We note that the GFSA layer discovers separate modes for break statements and regular control flow, and also learns to split probability mass across multiple trajectories in order to account for multiple paths through the program, closely following the program semantics. The paths learned by this policy are also quite long; the policy shown takes an average of 35 actions before accepting (on the 1x test set). More generally, this shows that the GFSA layer is able to learn many-hop reasoning that covers large distances in the graph by breaking down the reasoning into subcomponents defined by the learned automaton states.

## 5.3 Variable Misuse

Finally, we investigate performance on the variable misuse task [1, 39]. Following Hellendoorn et al. [19], we use a dataset of small code samples from a permissively-licenced subset of the ETH 150k Python dataset [33], where synthetic variable misuse bugs have been introduced in half of the examples by randomly replacing one of the identifiers with a different identifier in that program.[4] We train a model to predict the location of the incorrect identifier, as well as another location in the program containing the correct replacement that would restore the original program; we use a special "no-bug" location for the unmodified examples, similar to Vasic et al. [39] and Hellendoorn et al. [19].

We consider two graph neural network architectures: either an eight-layer RAT model [42] or eight GGNN blocks [27] with two message passing iterations per block (similar to Hellendoorn et al. [19]). For each, we investigate adding different types of edges to the base AST graph: no extra edges, hand-engineered edges used by Allamanis et al. [1] and Hellendoorn et al. [19], weighted

Table 2: Accuracy on the variable misuse task, in percent. "Start" indicates that edges are added to the base graph before running the graph model, and "middle" indicates they are added halfway through, conditioned on the output of the first half. Bold indicates overlapping 95% confidence intervals with the best model for each metric. See appendix D.3.3 for standard error estimates and additional details.

| Example type: | All | | No bug | | With bug | | | |
|---|---|---|---|---|---|---|---|---|
| Metric: | Full accuracy | | Classification | | Classification | | Loc & Repair | |
| Graph model family: | *RAT* | *GGNN* | *RAT* | *GGNN* | *RAT* | *GGNN* | *RAT* | *GGNN* |
| *Base AST graph only* | 88.22 | 83.52 | 92.05 | 91.26 | 93.03 | 88.15 | 88.30 | 81.63 |
| *Base AST graph, +2 layers* | 87.85 | 84.38 | 92.45 | 88.80 | 92.03 | 91.92 | 87.76 | 83.97 |
| *Hand-engineered edges* | 88.50 | 84.78 | 92.93 | 90.19 | 92.48 | 91.56 | 88.39 | 83.52 |
| *NRI head @ start* | 88.71 | 84.47 | 92.55 | 91.49 | 93.21 | 89.38 | 88.73 | 82.73 |
| *NRI head @ middle* | 88.42 | 84.41 | 92.83 | 88.29 | 92.31 | 92.20 | 88.62 | 84.44 |
| *Random walk @ start* | 88.91 | 84.52 | **93.22** | 91.35 | 92.77 | 89.28 | 88.73 | 82.96 |
| *RL ablation @ middle* | 87.28 | 84.96 | 90.36 | 90.44 | 93.71 | 90.64 | 87.73 | 84.30 |
| *GFSA layer (ours) @ start* | **89.47** | 85.01 | **93.10** | 90.08 | 93.56 | 91.80 | 89.58 | 83.91 |
| *GFSA layer (ours) @ middle* | **89.63** | 84.72 | 92.66 | 90.98 | **94.25** | 89.81 | **89.93** | 83.63 |

edges learned by a GFSA layer, weighted edges output by an NRI-like pairwise MLP, weighted edges produced by an ablation of GFSA consisting of a uniform random walk with a learned halting probability, and a single edge per start state sampled by a GFSA-based RL agent. For the NRI and GFSA layers, we investigate adding the edges either before the graph neural network model (building from the base graph), or halfway through the model (conditioned on the node embeddings from the first half). For the RL agent, we train with REINFORCE and a learned scalar reward baseline, and use the downstream cross-entropy loss as the reward. To show the effect of just increasing model capacity, we also present results for ten-layer models on the base graph. In all models, we initialize node embeddings based on a subword tokenization of the program (using the `Tensor2Tensor` library by Vaswani et al. [41]), and predict a joint distribution over the bug and repair locations, with softmax normalization and the standard cross entropy objective. See appendix D.3 for additional details on each of the above models, as well as results using an eight-layer GREAT model [19].

The results are shown in Table 2. We report overall accuracy, along with a breakdown by example type: for non-buggy examples, we report the fraction of examples the model predicts as non-buggy, and for buggy examples, we report both accuracy of the classification and accuracy of the predicted error and replacement identifier locations conditioned on the classification. Consistent with prior work, adding the hand-engineered features from Allamanis et al. [1] improves performance over only using the base graph. Interestingly, adding weighted edges using a random walk on the base graph yields similar performance to adding hand-engineered edges, suggesting that, for this task, improving connectivity may be more important than the specific program analyses used. We find that the GFSA layer combined with the RAT graph model obtains the best performance, outperforming the hand-engineered edges. Interestingly, we observe that the GFSA layer does not seem to converge to a discrete adjacency matrix, but instead assigns continuous weights. We conjecture that the output edge weights may provide additional representative power to the base model.

## 6 Conclusion

Inspired by ideas from programming languages and reinforcement learning, we propose the differentiable GFSA layer, which learns to add new edges to a base graph. We show that the GFSA layer can learn sophisticated behaviors for navigating grid-world environments and analyzing program behavior, and demonstrate that it can act as a viable replacement for hand-engineered edges in the variable misuse task. In the future, we plan to apply the GFSA layer to other domains and tasks, such as molecular structures or larger code repositories. We also hope to investigate the interpretability of the edges learned by the GFSA layer to determine whether they correspond to useful general concepts, which might allow the GFSA edges to be shared between multiple tasks.

## Broader Impact

We consider this work to be a general technical and theoretical contribution, without well-defined specific impacts. If applied to real-world program understanding tasks, extensions of this work might lead to reduced bug frequency or improved developer productivity. On the other hand, those benefits might accrue mostly to groups with sufficient resources to incorporate machine learning into their development practices. Additionally, if users put too much trust in the output of the model, they could inadvertently introduce bugs in their code because of incorrect model predictions. If applied to other tasks involving structured data, the impact would depend on the specific application; we leave the exploration of these other applications and their potential impacts to future work.

## Acknowledgments and Disclosure of Funding

We would like to thank Aditya Kanade and Charles Sutton for pointing out the connection to Reps [34], and Petros Maniatis for help with the variable misuse dataset. We would also like to thank Guillaume Rabusseau for advice regarding finite and weighted automata, Dibya Ghosh and Yujia Li for their helpful comments and suggestions during the writing process, and the Brain Learning for Code team at Google for useful feedback throughout the course of the project. Finally, we thank the reviewers for their feedback and for pointing out relevant related work. This work was entirely performed at and funded by Google.

## Footnotes

[1]For instance, the AST for `print(x + y)` contains nodes for `print`, `x`, `y`, `x + y`, and the call as a whole.

[2]An implementation is available at `https://github.com/google-research/google-research/tree/master/gfsa`.

[3]https://github.com/deepmind/labmaze

[4] https://github.com/google-research-datasets/great

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
