[Supplementary Material · appendix.pdf]

# A   Details on Constrained Reachability

In this section we describe how program analyses can be converted to regular languages, and provide the proofs for the statements in section 2.2.

## A.1   Example: Regular Languages For Program Analyses

The following grammar defines a regular language for the LastWrite edge type as described by Allamanis et al. [1], where $n_2$ is the last write to variable $n_1$ if there is a path from $n_1$ to $n_2$ whose label sequence matches the nonterminal **Last-Write**. We denote node type labels with capitals, edge type labels in lowercase, and nonterminal symbols in boldface. For simplicity we assume a single target variable name with its own node type TargetVariable, and only consider a subset of possible AST nodes.

| | | |
|---:|:--:|:---|
| | | *All* LastWrite *edges start from a use of the target variable.* |
| **Last-Write** | = | TargetVariable to-parent **Find-Current-Statement** |
| | | |
| | | *Once we find a statement, go backward.* |
| **Find-Current-Statement** | = | ExprStmt **Step-Backward** |
| | \| | Assign **Step-Backward** |
| | | *While in an expression, step out.* |
| | \| | BinOp to-parent **Find-Current-Statement** |
| | \| | Call to-parent **Find-Current-Statement** |
| | | |
| | | *Stop if we find an assignment to the target variable.* |
| **Check-Stmt** | = | Assign to-target TargetVariable |
| | | *Skip other statements.* |
| | \| | Assign to-target NonTargetVariable to-parent **Step-Backward** |
| | \| | ExprStmt **Step-Backward** |
| | | *Either enter If blocks or skip them.* |
| | \| | If to-last-child **Check-Stmt** |
| | \| | If **Step-Backward** |
| | | *Either enter While blocks or skip them, possibly jumping back to a break.* |
| | \| | While **Step-Backward** |
| | \| | While to-last-child **Check-Stmt** |
| | \| | While to-last-child **Find-Break-A** |
| | | |
| | | *If we have a previous statement, check it.* |
| **Step-Backward** | = | prev-stmt **Check-Stmt** |
| | | *If this is the first statement of an If block, exit.* |
| | \| | from-first-child If **Step-Backward** |
| | | *If this is the first statement of a While block, either exit or go back to the end of the loop body.* |
| | \| | from-first-child While **Step-Backward** |
| | \| | from-first-child While to-last-child **Check-Stmt** |
| | | |
| | | *If we find a Break, this is a possible previous loop exit point.* |
| **Find-Break-A** | = | Break **Step-Backward** |
| | | *Either way, keep looking for other break statements.* |
| | \| | Break **Find-Break-B** |
| | \| | ExprStmt **Find-Break-B** |
| | \| | If to-last-child **Find-Break-A** |
| | | *Don't enter while loops, since break statements only affect one loop.* |
| | \| | While **Find-Break-B** |
| | | |
| **Find-Break-B** | = | prev-stmt **Find-Break-A** |
| | \| | from-first-child If **Find-Break-B** |

The constructions for NextControlFlow and LastRead are similar. Note that for LastRead, instead of skipping entire statements until finding an assignment, the path must iterate over all expressions used within each statement and check for uses of the variable. For NextControlFlow, instead of stepping backward, the path steps forward, and instead of searching for break statements

after entering a loop, it searches for the containing loop when reaching a break statement. (This is because NEXTCONTROLFLOW simulates program execution forward instead of in reverse. Note that regular languages are closed under reversal [20], so such a transformation between forward and reverse paths is possible in general; we could similarly construct a language for LASTCONTROLFLOW if desired.)

## A.2 Proof of Proposition 1

We recall proposition 1:

**Proposition 1.** *Let $\mathcal{G}$ be a family of graphs annotated with node and edge types. There exists an encoding of graphs $G \in \mathcal{G}$ into POMDPs as described in section 3.1 and a mapping from regular languages $L$ into finite-state policies $\pi_L$ such that, for any $G \in \mathcal{G}$, there is an $L$-path from $n_0$ to $n_T$ in $G$ if and only if $p(a_T = \text{ADDEDGEANDSTOP}, n_T | n_0, \pi_L) > 0$.*

*Proof.* We start by defining a generic choice of POMDP conversion that depends only on the node and edge types. Let $G \in \mathcal{G}$ be a directed graph with node types $\mathcal{N}$, edge types $\mathcal{E}$, nodes $N$, and edges $E \subseteq N \times N \times \mathcal{E}$. We convert it to a POMDP by choosing $\Omega_{\tau(n)} = \{(\tau(n), \text{TRUE}), (\tau(n), \text{FALSE})\}$, $\mathcal{M}_{\tau(n)} = \mathcal{E}$,

$$p(n_{t+1}|n_t, a_t = (\text{MOVE}, m_t)) = \begin{cases} 1/|A_{n_t}^{m_t}| & \text{if } n_{t+1} \in A_{n_t}^{m_t}, \\ 1 & \text{if } n_{t+1} = n_t \text{ and } A_{n_t}^{m_t} = \varnothing, \\ 0 & \text{otherwise}, \end{cases}$$

$\omega_0 = (\tau(n_0), \text{TRUE})$, and $\omega_{t+1} = (\tau(n_{t+1}), n_{t+1} \in A_{n_t}^{m_t})$, where we let $A_{n_t}^{m_t} = \{n_{t+1} \mid (n_t, n_{t+1}, m_t) \in E\}$ be the set of neighbors adjacent to $n_t$ via an edge of type $m_t$.

Now suppose $L$ is a regular language over sequences of node and edge types. Construct a deterministic finite automaton $M$ that accepts exactly the words in $L$ (for instance, using the subset construction) [20]. Let $Q$ denote its state space, $q_0$ denote its initial state, $\delta : Q \times \Sigma \to Q$ be its transition function, and $F \subseteq Q$ be its set of accepting states. We choose $Q$ as the finite state memory of our policy $\pi_L$, i.e. at each step $t$ we assume our agent is associated with a memory state $z_t \in Q$. We let $z_0 = q_0$ be the initial memory state of $\pi_L$.

Consider an arbitrary memory state $z_t \in Q$ and observation $\omega_t = (\tau(n_t), e_t)$. We now construct a set of possible next actions and memories $N_t \subseteq \mathcal{A}_t \times Q$. If $e_t = \text{FALSE}$, let $N_t = \varnothing$. Otherwise, let $z_{t+1/2} = \delta(z_t, \tau(n_t))$. If $z_{t+1/2} \in F$, add $(\text{ADDEDGEANDSTOP}, z_{t+1/2})$ to $N_t$. Next, for each $m \in \mathcal{E}$, add $((\text{MOVE}, m), \delta(z_{t+1/2}, m))$ to $N_t$. Finally, let

$$\pi_L(a_t, z_{t+1}|z_t, \omega_t) = \begin{cases} 1/|N_t| & \text{if } (a_t, z_{t+1}) \in N_t, \\ 1 & \text{if } N_t = \varnothing, a_t = \text{STOP}, z_{t+1} = z_t \\ 0 & \text{otherwise}. \end{cases}$$

The $e_t = \text{FALSE} \implies N_t = \varnothing$ constraint ensures that the partial sequence of labels along any accepting trajectory matches the sequence of node type observations and movement actions produced by $\pi_L$. Since $\pi_L$ starts in the same state as $M$, and assigns nonzero probability to exactly the state transitions determined by $\delta$, it follows that the memory state of the agent along any partial trajectory $[n_0, m_0, n_1, m_1, \ldots, n_t]$ corresponds to the state of $M$ after processing the label sequence $[\tau(n_0), m_0, \tau(n_1), m_1, \ldots, \tau(n_t)]$.

Since $\pi_L$ assigns nonzero probability to the ADDEDGEANDSTOP action exactly when memory state is an accepting state from $F$, and $M$ is in an accepting state from $F$ exactly when the label sequence is in $L$, we conclude that desired property holds. $\square$

**Corollary.** *There exists an encoding of program AST graphs into POMDPs and a specific policy $\pi_{\text{NEXT-CF}}$ with finite-state memory such that $p(a_T = \text{ADDEDGEANDSTOP}, n_T \mid n_0, \pi) > 0$ if and only if $(n_0, n_T)$ is an edge of type NEXTCONTROLFLOW in the augmented AST graph. Similarly, there are policies $\pi_{\text{LAST-READ}}$ and $\pi_{\text{LAST-WRITE}}$ for edges of type LASTREAD and LASTWRITE, respectively.*

*Proof.* This corollary follows directly from Proposition 1 and the existence of regular languages for these edge types (see appendix A.1). $\square$

Note that equivalent policies also exist for POMDPs encoded differently than the proof of proposition 1 describes. For instance, instead of having "TargetVariable" as a node type and constructing edges for each target variable name separately, we can extend the observation $\omega_t$ to contain information on whether the current variable name matches the initial variable name and then find all edges at once, which we do for our experiments. Additionally, if an action would cause the policy to transition into an absorbing but non-accepting state (i.e. a failure state) in the discrete finite automaton for $L$, the policy can immediately take a BACKTRACK or STOP action instead, or reallocate probability to other states, instead of just cycling forever in that state. This allows the policy $\pi$ to more evenly allocate probability across possible answers, and we observe that the GFSA policies learn to do this in our experiments.

## B   Graph to POMDP Dataset Encodings

Here we describe the encodings of graphs as POMDPs that we use for our experiments.

### B.1   Python Abstract Syntax Trees

We convert all of our code samples into the unified format defined by the `gast` library,[5] which is a slightly-modified version of the abstract syntax tree provided with Python 3.8 that is backward-compatible with older Python versions. We then use a generic mechanism to convert each AST node into one or more graph nodes and corresponding POMDP states.

Each AST node type $\tau$ (such as `FunctionDef`, `If`, `While`, or `Call`) has a fixed set $F$ of possible field names. We categorize these fields into four categories: optional fields $F_{\text{opt}}$, exactly-one-child fields $F_{\text{one}}$, nonempty sequence fields $F_{\text{nseq}}$, and possibly empty sequence fields $F_{\text{eseq}}$. We define the observation space at nodes of type $\tau$ as

$$\Omega_\tau = \{\tau\} \times \Gamma \times \Psi_\tau$$

where $\Gamma$ is a task-specific extra observation space, and $\Psi_\tau$ indicates the result of the previous action:

$$\Psi_\tau = \{(\text{FROM}, f) \mid f \in F \cup \{\text{PARENT}\}\} \cup \{(\text{MISSING}, f) \mid f \in F_{\text{opt}} \cup F_{\text{eseq}}\}.$$

The $(\tau, \gamma, (\text{FROM}, f))$ observations are used when the agent moves to an edge of type $\tau$ from a child from field $f$ (or from the parent node), and the $(\tau, \gamma, (\text{MISSING}, f))$ observations are used when the agent attempts to move to a child for field $f$ but no such child exists. We define the movement space as

$$\mathcal{M}_\tau = \{\text{GO-PARENT}\} \cup \{(\text{GO}, f) \mid f \in F_{\text{one}} \cup F_{\text{opt}}\}$$
$$\cup \{(x, f) \mid x \in \{\text{GO-FIRST}, \text{GO-LAST}, \text{GO-ALL}\}, f \in F_{\text{nseq}} \cup F_{\text{eseq}}\}.$$

GO moves the agent to the single child for that field, GO-FIRST moves it to the first child, GO-LAST moves it to the last child, and GO-ALL distributes probability evenly among all children. GO-PARENT moves the agent to the parent node; we omit this movement action for the root node ($\tau = \text{Module}$).

For each sequence field $f \in F_{\text{nseq}} \cup F_{\text{eseq}}$, we also define a helper node type $\tau_f$, which is used to construct a linked list of children. This helper node has the fixed observation space

$$\Psi_{\tau_f} = \{\text{FROM-PARENT}, \text{FROM-ITEM}, \text{FROM-NEXT}, \text{FROM-PREV},$$
$$\text{MISSING-NEXT}, \text{MISSING-PREV}\}$$

and action space

$$\mathcal{M}_{\tau_f} = \{\text{GO-PARENT}, \text{GO-ITEM}, \text{GO-NEXT}, \text{GO-PREV}\}.$$

When encoding the AST as a graph, helper nodes of this type are inserted between the AST node of type $\tau$ and the children for field $f$: the "parent" of a helper node is the original AST node, and the "item" of the $n$th helper node is the $n$th child of the original AST node for field $f$.

We note that this construction differs from the construction in the proof of proposition 1, in that movement actions are specific to the node type of the current node. When the agent takes the GO-PARENT action, the observation for the next step informs it what field type it came from. This helps

keep the state space of the GFSA policy small, since it does not have to guess what its parent node is and then remember the results; it can instead simply walk to the parent node and then condition its next action on the observed field. The construction described here still allows encoding the edges from NEXTCONTROLFLOW, LASTREAD, and LASTWRITE as policies, as we empirically demonstrate by training the GFSA layer to replicate those edges.

## B.2 Grid-world Environments

For the grid-world environments, we represent each traversable grid cell as a node, and classify the cells into eleven node types corresponding to which movement directions (left, up, right, and down) are possible:

$$\mathcal{N} = \{\mathrm{LU}, \mathrm{LR}, \mathrm{LD}, \mathrm{UR}, \mathrm{UD}, \mathrm{RD}, \mathrm{LUR}, \mathrm{LUD}, \mathrm{LRD}, \mathrm{URD}, \mathrm{LURD}\}$$

Note that in our dataset, no cell has fewer than two neighbors.

For each node type $\tau \in \mathcal{N}$ the movement actions $\mathcal{M}_\tau$ correspond exactly to the possible directions of movement; for instance, cells of type LD have $\mathcal{M}_{\mathrm{LD}} = \{\mathrm{L}, \mathrm{D}\}$. We use a trivial observation space $\Omega_\tau = \{\tau\}$, i.e. the GFSA automaton sees the type of the current node but no other information.

When converting grid-world environments into POMDPs, we remove the BACKTRACK action to encourage the GFSA edges to match more traditional RL option sub-policies.

# C  GFSA Layer Implementation

Here we describe additional details about the implementation of the GFSA layer.

## C.1  Parameters

We represent the parameters $\theta$ of the GFSA layer as a table indexed by feasible observation and action pairs $\Phi$ as well as state transitions:

$$\Phi = \left( \bigcup_{\tau \in \mathcal{N}} \Omega_\tau \times \mathcal{A}_\tau \right), \qquad \theta : Z \times \Phi \times Z \to \mathbb{R},$$

where $Z = \{0, 1, \ldots, |Z|-1\}$ is the set of memory states. We treat the elements of $\theta$ as unnormalized log-probabilities and then set $\pi = \mathrm{softmax}(\theta)$, normalizing separately across actions and new memory states for each possible current memory state and observation.

To initialize $\theta$, we start by defining a "base distribution" $p$, which chooses a movement action at random with probability 0.95 and a special action (ADDEDGEANDSTOP, STOP, BACKTRACK) otherwise, and which stays in the same state with probability 0.8 and changes states randomly otherwise. Next, we sample our initial probabilities $q$ from a Dirichlet distribution centered on $p$ (with concentration parameters $\alpha_i = p_i/\beta$ where $\beta$ is a temperature parameter), and then take a (stabilized) logarithm $\theta_i = \log(q_i + 0.001)$. This ensures that the initial policy has some initial variation, while still biasing it toward staying in the same state and taking movement actions most of the time.

## C.2  Algorithmic Details

As a preprocessing step, for each graph in the dataset, we compute the set $X$ of all $(n, \omega)$ node-observation pairs for the corresponding MDP. We then compute "template transition matrices", which specify how to convert the probability table $\theta$ into a transition matrix by associating transitions $X \times X$ and halting actions $X \times \{\mathrm{ADDEDGEANDSTOP}, \mathrm{STOP}, \mathrm{BACKTRACK}\}$ with their appropriate indicies into $\Phi$. Then, when running the model, we retrieve blocks of $\theta$ according to those indices to construct the transition matrix for that graph (implemented with "gather" and "scatter" operations).

Conceptually, each possible starting node $n_0$ could produce a separate transition matrix $Q_{n_0}$ : $X \times Z \times X \times Z \to \mathbb{R}$ because part of the observation in each state (which we denote $\gamma \in \Gamma$ and leave out of $X$) may depend on the starting node or other learned parameters. We address this by instead computing an "observation-conditioned" transition tensor

$$Q : \Gamma \times X \times Z \times X \times Z \to \mathbb{R}$$

that specifies transition probabilities for each observation $\gamma$, along with a start-node-conditioned observation tensor

$$C : N \times X \times \Gamma \rightarrow \mathbb{R}$$

that specifies the probability of observing $\gamma$ for a given start node $n_0$ and current $(n, \omega)$ tuple. In order to compute a matrix-vector product $Q_{n_0} \boldsymbol{v}$ we can then use the tensor product

$$\sum_{i,z,\gamma} C_{n_0,i,\gamma} \, Q_{\gamma,i,z,i',z'} \, v_{i,z}$$

which can be computed efficiently without having to materialize a separate transition matrix for every start node $n_0 \in N$.

During the forward pass through the GFSA layer, to solve for the absorbing probabilities in equation 1, we iterate

$$\boldsymbol{x}_0 = \boldsymbol{\delta}_{n_0}, \qquad\qquad\qquad \boldsymbol{x}_{k+1} = \boldsymbol{\delta}_{n_0} + Q_{n_0} \boldsymbol{x}_k$$

until a fixed number of steps $K$, then approximate

$$p(a_T, n_T | n_0, \pi) = H_{(a_T, n_T),:} \, (I - Q_{n_0})^{-1} \, \boldsymbol{\delta}_{n_0} \approx H_{(a_T, n_T),:} \, \boldsymbol{x}_K.$$

To efficiently compute the backwards pass without saving all of the values of $\boldsymbol{x}_k$, we use the `jax.lax.custom_linear_solve` function from JAX [8], which converts the gradient equations into a transposed matrix system

$$\left(I - Q_{n_0}^T\right)^{-1} H^T \frac{\partial \mathcal{L}}{\partial p(\cdot | n_0, \pi)}$$

that we similarly approximate with

$$\boldsymbol{y}_0 = H^T \frac{\partial \mathcal{L}}{\partial p(\cdot | n_0, \pi)}, \qquad\qquad \boldsymbol{y}_{k+1} = H^T \frac{\partial \mathcal{L}}{\partial p(\cdot | n_0, \pi)} + Q_{n_0}^T \boldsymbol{y}_k.$$

Note that both iteration procedures are guaranteed to converge because the matrix $I - Q_{n_0}$ is diagonally dominant [35]. Conveniently, we implement $Q_{n_0} \boldsymbol{x}_k$ using the tensor product described above, and JAX automatically translates this into a computation of the transposed matrix-vector product $Q_{n_0}^T \boldsymbol{y}_k$ using automatic differentiation.

If the GFSA policy assigns a very large probability to the BACKTRACK action, this can lead to numerical instability when computing the final adjacency matrix, since we condition on non-backtracking trajectories when computing our final adjacency matrix. We circumvent this issue by constraining the policy such that a small fraction of the time ($\varepsilon_{\text{bt-stop}}$), if it attempts to take the BACKTRACK action, it instead takes the STOP action; this ensures that, if the policy backtracks with high probability, the weight of the produced edges will be low. Additionally, we attempt to mitigate floating-point precision issues during normalization by summing over ADDEDGEANDSTOP and STOP actions instead of computing $1 - p(a_t = \text{BACKTRACK} | \cdots)$ directly.

When computing an adjacency matrix from the outputs of the GFSA layer, there are two ways to extract multiple edge types. The first is to associate each edge type with a distinct starting state in $Z$. The second way is to compute a different version of the parameter vector $\theta$ for each edge type. For the variable misuse experiments, we use the first method, since sharing states uses less memory. For the grid-world experiments, we use the second method, as we found that using non-shared states gives slightly better performance and results in more interpretable learned options.

## C.3  Asymptotic Complexity

We now give a brief complexity analysis for the GFSA layer implementation. Let $n$ be the number of nodes, $e$ be the number of edges, $z$ be the number of memory states, $\omega$ be the number of "static" observations per node (such as FROM-PARENT), and $\gamma$ be the number of "dynamic" observations per node (for instance, observations conditioned on learned node embeddings).

**Memory:** Storing the probability of visiting each state in $X \times Z$ for a given source node takes memory $O(n\omega z)$, so storing it for all source nodes takes $O(n^2 \omega z)$. For a dense representation of $Q$ and $C$, $Q$ takes $O(n^2 \omega^2 z^2 \gamma)$ memory and $C$ takes $O(n^2 \omega \gamma)$. Since memory usage is independent of

the number of iterations, the overall memory cost is thus $O(n^2\omega^2z^2\gamma)$. We note that memory scales proportional to the square of the number of nodes, but this is in a sense unavoidable since the output of the GFSA layer is a dense $n \times n$ matrix even if $Q$ is sparse.

**Time:** Computing the tensor product for all starting nodes requires computing $n \times n\omega z \times n\omega z \times \gamma$ elements. Since we do this at every iteration, we end up with a time cost of $O(T_{\max} n^3\omega^2z^2\gamma)$. We note that a sparse representation of $Q$ might reduce this to $O(T_{\max}(n^2z + nez^2)\omega\gamma)$ (since we could first contract with C with cost $n \times n\omega \times z \times \gamma$ and then iterate over edges, start nodes, observations, and states, with cost $n \times e \times \omega z \times z \times \gamma$). However, in practice we use a dense implementation to take advantage of fast accelerator hardware.

# D    Experiments, Hyperparameters, and Detailed Results

Here we describe additional details for each of our experiments and the corresponding evaluation results. For all of our experiments, we train and evaluate on TPU v2 accelerators.[6] Each training job uses 8 TPU v2 cores, evenly dividing the batch size between the cores and averaging gradients across them at each step.

## D.1    Grid-world Task

We use the LabMaze generator (`https://github.com/deepmind/labmaze`) from DeepMind Lab [6] to generate our grid-world layouts. We configure it with a width and height of 19 cells, a maximum of 6 rooms, and room sizes between 3 and 6 cells on each axis. We then convert the generated grids into graphs, and filter out examples with more than 256 nodes or 512 node-observation tuples. We generate 100,000 training graphs and 100,000 validation graphs. For each graph, we then pick 32 goal locations uniformly at random.

We configure the GFSA layer to use four independent policies to produce four derived edge types. For each policy, we set the memory space to the two-element set $Z = \{0, 1\}$, where $z_0 = 0$. We initialize parameters using temperature $\beta = 0.2$, but use the Dirichlet sample directly as a logit, i.e. $\theta_i = q_i$. (We found that applying the logarithm from appendix C during initialization yields similar numerical performance but makes the learned policies harder to visualize.)

Since the interpretation of the edges as options requires them to be properly normalized (i.e. the distribution $p(s_{t+1}|s_t, a_t)$ must be well defined), we make a few modifications to the output adjacency matrix produced by the GFSA layer. In particular, we do not use the learned adjustment parameters described in section 3.3, instead fixing $a = 1, b = 0$. We also ensure that the edge weights are normalized to 1 for each source node by assigning any missing mass to the diagonal. In other words, if the GFSA sub-policy agent takes a STOP action or fails to take the ADDEDGEANDSTOP action before $T_{\max}$ iterations, we instead treat the option as a no-op that causes the primary agent to remain in place. We also remove the BACKTRACK action.

At each training iteration, we sample a graph $G$ from our training set, and in parallel compute approximate entropy-regularized optimal policies $\pi^*$ for each of the 32 goal locations for $G$. Mathematically, for each goal $g$, we seek

$$\pi^* = \operatorname*{argmax}_{\pi} \mathbb{E}_{(s_t, a_t) \sim p(\cdot \mid \pi)} \left[ \sum_{0 \le t \le T} -1 + \mathcal{H}(\pi(\cdot \mid s_t)) \,\middle|\, s_T = g \right],$$

where $\mathcal{H}(\pi(\cdot \mid s_t))$ denotes the entropy of the distribution over actions, and we have fixed the reward to -1 for all timesteps. We use an entropy-regularized objective here so that the policy $\pi^*$ is nondeterministic and thus has a useful derivative with respect to the option distribution. As described by Haarnoja et al. [18], we can compute this optimal policy by doing soft-Q iteration using the update equations

$$Q_{\text{soft}}(s_t, a_t) \leftarrow \mathbb{E}_{s_{t+1} \sim p(\cdot|s_t, a_t)} \left[ V_{\text{soft}}(s_{t+1}) \right] - 1,$$

$$V_{\text{soft}}(s_t) \leftarrow \log \sum_{a_t} \exp\left( Q_{\text{soft}}(s_t, a_t) \right).$$

until reaching a fixed point, and then letting

$$\pi^*(a_t|s_t) = \exp(Q_{\text{soft}}(s_t, a_t) - V_{\text{soft}}(s_t)).$$

Since our graphs are small, we can store $Q_{\text{soft}}$ and $V_{\text{soft}}$ in tabular form, and directly solve for their optimal values by iterating the above equations. In practice, we approximate the solution by using 512 iterations.

After computing $Q_{\text{soft}}^{(g)}$, $V_{\text{soft}}^{(g)}$, and $\pi^*_{(g)}$ for each choice of $g$, we then define a minimization objective for the full task as

$$\mathcal{L} = -\mathbb{E}_{s_0, g}\left[V_{\text{soft}}^{(g)}(s_0)\right],$$

i.e. we seek to maximize the soft value function across randomly chosen sources and goals, or equivalently to minimize the expected number of steps taken by $\pi^*_{(g)}$ before reaching the goal. We compute gradients by using implicit differentiation twice: first to differentiate through the fixed point to the soft-Q iteration, and second to differentiate through the computation of the GFSA edges. Implicitly differentiating through the soft-Q equations is conceptually similar to implicit MAML [32] except that the parameters we optimize in the outer loop (the GFSA parameters) are not the same as the parameters we optimize in the inner loop (the graph-specific tabular policy).

Note that differentiating through the soft-Q fixed point requires first linearizing the equations around the fixed point. More specifically, if we express the fixed point equations in terms of a function $V_{\text{soft}} = f(V_{\text{soft}}, \theta)$ where $\theta$ represents the GFSA parameters, we have

$$\partial V_{\text{soft}} = \partial f(V_{\text{soft}}, \theta) = f_V(V_{\text{soft}}, \theta)\partial V_{\text{soft}} + f_\theta(V_{\text{soft}}, \theta)\partial\theta$$

(where $f_V(V_{\text{soft}}, \theta)$ denotes the Jacobian of $f$ with respect to $V$, and similarly for $f_\theta$) and thus

$$\partial V_{\text{soft}} = \left(I - f_V(V_{\text{soft}}, \theta)\right)^{-1} f_\theta(V_{\text{soft}}, \theta)\partial\theta$$

which leads to gradient equations

$$\frac{\partial\mathcal{L}}{\partial\theta} = f_\theta(V_{\text{soft}}, \theta)^T \left(I - f_V(V_{\text{soft}}, \theta)^T\right)^{-1} \frac{\partial\mathcal{L}}{\partial V_{\text{soft}}}.$$

As before, JAX makes it possible to easily express these gradient computations and automatically handles the computation of the relevant partial derivatives and Jacobians. In this case, due to the small number of goal locations and lack of diagonal dominance guarantees, we simply compute and invert the matrix $I - f_V(V_{\text{soft}}, \theta)^T$ during the backward pass instead of using an iterative solver. (See Liao et al. [28] for additional information about implicitly differentiating through fixed points.)

We trained the model using the Adam optimizer, with a learning rate of 0.001 and a batch size of 32 graphs with 32 goals each, for approximately 50,000 iterations, until the validation loss plateaued. We then picked a grid from the validation set, and chose four possible starting locations manually to give a summary of the overall learned behavior.

## D.2 Static Analyses

### D.2.1 Datasets

For the static analysis tasks, we first generate a dataset of random Python programs using a probabilistic context free grammar. This grammar contains a variety of nonterminals and associated production rules:

- **Number**: An integer or float expression. Either a variable dereference, a constant integer, an arithmetic operation, or a function call with numeric arguments.
- **Boolean**: A boolean expression. Either a comparison between numbers, a constant `True` or `False`, or a boolean combination using `and` or `or`.
- **Statement**: A single statement. Either an assignment, a call to `print`, an if, if-else, for, or while block, or a `pass` statement.
- **Block**: A contiguous sequence of statements that may end in a `return`, `break`, or `continue`, or with a normal statement; we only allow these statements at the end of a block to avoid producing dead code.

```
def generated_function(a, b):
    for v2 in range(int(bar_2(a, b))):
        v3 = foo_4(v2, b, bar_1(b), a) / 42
        v3 = (b + 8) * foo_1((v2 * v3))
        pass
        v3 = b
        while False:
            a = v3
            a = v3
            v2 = 34
            break
        if bar_1((b * b)) != v2:
            v4 = foo_4(bar_2(56, bar_1(v2)),
                       foo_4(b, a, a, 39) - v2,
                       bar_1(a), 32)
            for v5 in range(int(v4)):
                v6 = v4
                pass
                break
            print(69)
        v2 = v2
    b = 15
    b = ((a + 96) + 89) - a
    v2 = foo_4(b, 21, 26, foo_4(85, a, a - b, a))
    b = v2 - (a - v2)
```

Figure D.1: Example of a program from the "1x" program distribution.

```
def generated_function(a, b):
    if bar_1(b) > b:
        b = a
        print(b)
    else:
        a = a + a
    a = bar_1(62 - 35)
    if b <= bar_1(54):
        b = b
        while a >= 58:
            b = foo_1(a)
            pass
            pass
    else:
        a = bar_4(b, bar_1(b), bar_1(a * a), bar_1(a))
    b = 88
```

Figure D.2: Example of a program from the "0.5x" program distribution.

```
def generated_function(a, b):
    v2 = b
    pass
    b = v2
    pass
    v2 = b
    b = bar_1(v2)
    v3 = v2
    print(b)
    b = bar_1(v3) + (bar_1(20) - v2)
    print(56)
    if (foo_2(v2 + v3, foo_1(a)) == a) or ((foo_2(b, v2) < 22) or (v2 >= a or 37 <= v3)):
        v3 = b
        print(v2)
        print(foo_1(a))
        a = v3
        b = foo_1(a)
        v4 = foo_1(b)
        print(foo_1(v3) * bar_1(v2))
        b = a
        print(bar_1(v2))
        v2 = v2
        v4 = 67
        v5 = bar_2((v4 + v2) / (a / b), b)
    else:
        v4 = v3
        b = v4
        while ((v2 + (a / v4)) * (foo_2(93, v2) + v2)) < ((a - v2) - 18):
            v5 = v2
            v6 = bar_2(v3, (a - v4) + v3)
            a = v2 + v2
            break
        v5 = 71 / v2
        v6 = (a + (b + 47)) - (foo_2(v2, a) / (v5 * v3))
        b = v5
        b = foo_2(v3, v4)
        v5 = 14
        v3 = v3
    v4 = b * foo_1(b)
    v5 = bar_4(v2, v3, v4, b)
    v6 = a
    v3 = v5
    b = 11
    v7 = foo_1(v2)
    v8 = v4
    v4 = foo_1(foo_1(b))
    v4 = bar_1(bar_1(bar_2(32, v5)))
    v5 = bar_1(bar_2(v2, v2))
```

Figure D.3: Example of a program from the "2x" program distribution.

We apply constraints to the generation process such that variable names are only used after they have been defined, expressions are limited to a maximum depth, and statements continue to be generated until reaching a target number of AST nodes. For the training dataset, we set this target number of nodes to 150, and convert each generated AST into a graph according to B.1; we then throw out graphs with more than 256 graph nodes or 512 node-observation tuples. For our test datasets, we use a target AST size of 300 AST nodes and cutoffs of 512 graph nodes or 1024 node-observation tuples for the "2x" dataset, and a target of 75 AST nodes and cutoffs of 128 graph nodes and 512 tuples for the "0.5x" dataset. For each dataset, the graph size cutoff results in keeping approximately 95% of the generated ASTs. Figures D.1, D.2, and D.3 show example programs from these distributions.

We generated a training dataset with 100,000 programs, a validation dataset of 1024 programs, and a test dataset of 100,000 programs for each of the three sizes (1x, 0.5x, 2x).

### D.2.2 Architectures and Hyperparameters

We configure the GFSA layer to produce a single edge type, corresponding to the target edge of interest. For this task, we specify the the task-specific observation $\gamma$ referenced in appendix B.1 such that the agent can observe when its current node is a variable with the same identifier as the initial node. We treat $|Z|$ as a hyperparameter, varying between 2, 4, and 8, with a fixed starting state $z_0$. We additionally randomly sample the backtracking stability hyperparameter $\varepsilon_{\text{bt-stop}}$ according to a log-uniform distribution within the range $[0.001, 0.1]$ (see appendix C). We initialize parameters with

temperature $\beta = 0.01$. Since we choose an optimal threshold while computing the F1 score, we do not use the learned adjustment parameters described in section 3.3, and instead fix $a = 1, b = 0$.

For the GGNN, GREAT, and RAT baselines, we evaluate with both "nodewise" and "dot-product" heads. For the "nodewise" head, we compute outputs as

$$A_{n,n'} = \sigma\big(\big[f_\theta(X_{\text{node}} + \boldsymbol{b}^T \boldsymbol{\delta}_n, X_{\text{edge}})\big]_{n'}\big)$$

where the learned model $f_\theta : \mathbb{R}^{d \times |N|} \times \mathbb{R}^{e \times |N| \times |N|} \to \mathbb{R}^{|N|}$ produces a scalar output for each node, $X_{\text{node}} \in \mathbb{R}^{d \times |N|}$ and $X_{\text{edge}} \in \mathbb{R}^{e \times |N| \times |N|}$ are embeddings of the node and edge features, $\boldsymbol{\delta}_n$ is a one-hot vector indicating the start node, and $\boldsymbol{b}$ is a learned start node embedding. For the "dot-product" head, we instead compute

$$Y = f_\theta(X_{\text{node}}, X_{\text{edge}}), \qquad\qquad A_{n,n'} = \sigma\big(\boldsymbol{y}_n^T W \boldsymbol{y}_{n'} + b\big),$$

where the learned model $f_\theta : \mathbb{R}^{d \times |N|} \times \mathbb{R}^{e \times |N| \times |N|} \to \mathbb{R}^{d \times |N|}$ produces updated node embeddings $\boldsymbol{y}_n$, $W$ is a learned $d \times d$ matrix, and $b$ is a learned scalar bias. Since the nodewise models require $|N|$ times as many more forward passes to compute edges for a single example, we keep training time manageable by reducing the width relative to the dot-product models.

The RAT and GREAT models are both variants of a transformer applied to the nodes of a graph. Both models use a set of attention heads, each of which compute query and key vectors $\boldsymbol{q}_n, \boldsymbol{k}_n \in \mathbb{R}^d$ for each node $n$ as linear transformations of the node features $\boldsymbol{x}_n$: $\boldsymbol{q}_n = W^Q \boldsymbol{x}_n$, $\boldsymbol{k}_n = W^K \boldsymbol{x}_n$. The RAT model computes attention logits as

$$y_{(n,n')} = \frac{\boldsymbol{q}_n^T\big(\boldsymbol{k}_{n'} + W^{EK} \boldsymbol{e}_{(n,n')}\big)}{\sqrt{d}}$$

where we transform the edge features $\boldsymbol{e}_{(n,n')}$ into an "edge key" that can be attended to by the query in addition to the content-based key. This corresponds to the attention equations as described by Shaw et al. [37], but with a graph-based mechanism for choosing the pairwise key vector. The GREAT model uses an easier-to-compute formulation

$$y_{(n,n')} = \frac{\boldsymbol{q}_n^T \boldsymbol{k}_{n'} + \boldsymbol{w}^T \boldsymbol{e}_{(n,n')} \cdot \mathbf{1}^T \boldsymbol{k}_{n'}}{\sqrt{d}}$$

where the attention logits are biased by a (learned) linear projection of the edge features, scaled by a (fixed) linear projection of the key ($\mathbf{1}$ denotes a vector of ones). In both models, the $y_{(n,n')}$ are converted to attention weights $\alpha_{(n,n')}$ using softmax, and used to compute a weighed average of embedded values. However, in the RAT model, both nodes and edges contribute to values ($z_n = \sum_{n'} \alpha_{(n,n')}(\boldsymbol{v}_{n'} + W^{EV} \boldsymbol{e}_{(n,n')})$), whereas in GREAT this sum is only over nodes ($z_n = \sum_{n'} \alpha_{(n,n')} \boldsymbol{v}_{n'}$).

For the NRI-encoder-based model, we make multiple adjustments to the formulation from Kipf et al. [25] in order to apply it to our setting. Since we are adding edges to an existing graph, the first part of our NRI model combines aspects from the encoder and decoder described in Kipf et al. [25]; we express our version in terms of blocks that each compute

$$\boldsymbol{h}_{(n,n')}^{i+1} = \sum_k e_{k,(n,n')} f_e^{i,k}(\boldsymbol{h}_n^i, \boldsymbol{h}_{n'}^i), \qquad\qquad \boldsymbol{h}_n^{i+1} = f_v^i\left(\sum_{n'} \boldsymbol{h}_{(n,n')}^{i+1}\right).$$

where $\boldsymbol{h}_n^i$ denotes the vector of node features after layer $i$, $\boldsymbol{h}_{(n,n')}^{i+1}$ denotes the vector of hidden pairwise features, and $e_{k,(n,n')}$ is the $k$th edge feature between $n$ and $n'$ from the base graph. To enable deeper models, we apply layer normalization and residual connections after each of these blocks, as in Vaswani et al. [40]. We then compute the final output head by applying the sigmoid activation to the final layer's hidden pairwise feature matrix $\boldsymbol{h}_{(n,n')}^I$ (which we constrain to have feature dimension 1), replacing the softmax used in the original NRI encoder (since we are doing binary classification, not computing a categorical latent variable). All versions of $f$ are learned MLPs with ReLU activations.

The RL agent baseline uses the same parameterization as the GFSA layer. However, instead of exactly solving for marginals, we sample a discrete transition at every step. Given a particular start node, the

agent gets a reward of +1 if it takes the ADDEDGEANDSTOP action at any of the correct destination nodes, or if it takes the STOP action and there was no correct destination node. We use 20 rollouts per start node, and train with REINFORCE and a leave-one-out control variate. During final evaluation, we compute exact marginals as for the GFSA layer; thus, differences in evaluation results reflect differences in the learning algorithm only.

For all of our baselines, we convert the Python AST into a graph by transforming the AST nodes into graph nodes and the field relationships into edges. For parity with the GFSA layer, the helper nodes defined in appendix B.1 are also used in the the baseline graph representation, and we add an extra edge type connecting variables that use the same identifier. All edges are embedded in both forward and reverse directions. We include hyperparameters for whether the initial node embeddings $X_{\text{node}}$ contain positional encodings computed as in Vaswani et al. [40] according to a depth-first tree traversal, and whether edges are embedded using a learned vector or using a one-hot encoding.

For the GGNN model, we choose a number of GGNN iterations (between 4 and 12 iterations using the same parameters) and a hidden state dimension (from $\{16, 32, 128\}$ for the nodewise models or $\{128, 256, 512\}$ for the dot-product models).

For the GREAT and RAT models, we choose a number of layers (between 4 and 12, but not sharing parameters), a hidden state dimension (from $\{16, 32, 128\}$ for the nodewise models or $\{128, 256, 512\}$ for the dot-product models), and a number of self-attention heads (from $\{2, 4, 8, 16\}$), with query, key, and value sizes chosen so that the sum of sizes across all heads matches the hidden state dimension.

For the NRI encoder model, we choose whether to allow communication between non-adjacent nodes, a hidden size for node features (from $\{128, 256, 512\}$), a hidden size for intermediate pairwise features (from $\{16, 32, 64\}$), a hidden size for initial base-graph edge features (from $\{16, 32, 64\}$), a depth for each MLP (from 1 to 5 layers), and a number of NRI-style blocks (between 4 and 12).

### D.2.3  Training and Detailed Results

For all of our models, we train using the Adam optimizer for either 500,000 iterations or 24 hours, whichever comes first; this is enough time for all models to converge to their final accuracy. For each model version and task, we randomly sample 32 hyperparameter settings, and then select the model and early-stopping point with the best F1 score on a validation set of 1024 functions. In addition to the hyperparameters described above, all models share the following hyperparameters: batch size (either 8, 32, or 128), learning rate (log-uniform in $[10^{-5}, 10^{-2}]$), gradient clipping threshold (log-uniform in $[1, 10^4]$), and focal-loss temperature $\gamma$ (uniform in $[0, 5]$). Hyperparameter settings that result in out-of-memory errors are not counted toward the 32 samples.

After selecting the best performing model for each model type and task based on performance on the validation set, we evaluated the model on each of our test datasets. For each example size (1x, 2x, 0.5x), we partitioned the 100,000 test examples into 10 equally-sized folds. We used the first fold to tune the final classifier threshold to maximize F1 score (using a different threshold for each example size to account for shifts in the distribution of model outputs). We then fixed that threshold and evaluated the F1 score on each of the other splits. We report the mean of the F1 score across those folds, along with an approximate standard error estimate (computed by dividing the standard deviation of the F1 score across folds by $\sqrt{9} = 3$).

To assess robustness of convergence, we also compute the fraction of training runs that achieve at least 90% accuracy on the validation set. Note that each training job has different hyperparameters but also a different parameter initialization and a different dataset iteration order; we do not attempt to distinguish between these sources of variation.

Table D.1 contains higher-precision results for the edge-classification tasks, along with the standard error estimates computed as above. Additionally, figure D.4 shows precision-recall curves, computed for a subset of the experiments that shows the most interesting variation in performance.

Table D.1: Full-precision results on static analysis tasks. Expressed as mean F1 score (in %) ± standard error on test set. For 1x dataset size, we also report fraction of training jobs across hyperparameter sweep that achieved 90% validation accuracy.

| Task | Next Control Flow | | |
|---|---|---|---|
| Example size | 1x | 2x | 0.5x |
| **100,000 training examples** | | | |
| *RAT nw* | 99.9837 ± 0.0006 (25/32 @ 90%) | 99.9367 ± 0.0012 | 99.9880 ± 0.0007 |
| *GREAT nw* | 99.9770 ± 0.0011 (26/32 @ 90%) | 99.8709 ± 0.0013 | 99.9834 ± 0.0010 |
| *GGNN nw* | 99.9823 ± 0.0007 (31/32 @ 90%) | 93.9034 ± 0.0304 | 97.7723 ± 0.0246 |
| *RAT dp* | 99.9945 ± 0.0004 (26/32 @ 90%) | 92.5278 ± 0.0080 | 96.5901 ± 0.0150 |
| *GREAT dp* | 99.9941 ± 0.0006 (24/32 @ 90%) | 96.3243 ± 0.0092 | 98.3557 ± 0.0081 |
| *GGNN dp* | 99.9392 ± 0.0014 (26/32 @ 90%) | 62.7524 ± 0.0195 | 98.5104 ± 0.0176 |
| *NRI encoder* | 99.9765 ± 0.0010 (31/32 @ 90%) | 85.9087 ± 0.0156 | 99.9161 ± 0.0021 |
| *RL ablation* | 94.2419 ± 0.0118 (02/32 @ 90%) | 93.5616 ± 0.0087 | 94.8329 ± 0.0241 |
| *GFSA Layer (ours)* | **99.9972 ± 0.0001** (29/32 @ 90%) | **99.9941 ± 0.0002** | **99.9985 ± 0.0002** |
| **100 training examples** | | | |
| *RAT nw* | 98.6324 ± 0.0090 (13/32 @ 90%) | 95.9320 ± 0.0092 | 96.3167 ± 0.0249 |
| *GREAT nw* | 98.2327 ± 0.0054 (13/32 @ 90%) | 97.9814 ± 0.0071 | 98.5181 ± 0.0065 |
| *GGNN nw* | 99.3749 ± 0.0060 (25/32 @ 90%) | 98.3590 ± 0.0050 | 98.6022 ± 0.0141 |
| *RAT dp* | 81.8068 ± 0.0296 (00/32 @ 90%) | 68.4592 ± 0.0187 | 87.0517 ± 0.0334 |
| *GREAT dp* | 86.5967 ± 0.0216 (00/32 @ 90%) | 62.9828 ± 0.0245 | 80.5810 ± 0.0192 |
| *GGNN dp* | 76.8530 ± 0.0388 (00/32 @ 90%) | 22.9947 ± 0.0083 | 28.9142 ± 0.0520 |
| *NRI encoder* | 81.7358 ± 0.0347 (00/32 @ 90%) | 69.0823 ± 0.0216 | 88.8749 ± 0.0452 |
| *RL ablation* | 91.6981 ± 0.0122 (03/32 @ 90%) | 91.1424 ± 0.0120 | 92.2917 ± 0.0215 |
| *GFSA Layer (ours)* | **99.9944 ± 0.0002** (29/32 @ 90%) | **99.9890 ± 0.0003** | **99.9971 ± 0.0004** |

| Task | Last Read | | |
|---|---|---|---|
| Example size | 1x | 2x | 0.5x |
| **100,000 training examples** | | | |
| *RAT nw* | 99.8602 ± 0.0020 (11/32 @ 90%) | 96.2865 ± 0.0083 | **99.9785 ± 0.0008** |
| *GREAT nw* | 99.9099 ± 0.0015 (14/32 @ 90%) | 95.1157 ± 0.0100 | **99.9801 ± 0.0006** |
| *GGNN nw* | 95.5197 ± 0.0121 (04/32 @ 90%) | 9.2216 ± 0.0658 | 86.2371 ± 0.0310 |
| *RAT dp* | 99.9579 ± 0.0011 (18/32 @ 90%) | 42.5754 ± 0.0139 | 91.9595 ± 0.0325 |
| *GREAT dp* | **99.9869 ± 0.0005** (18/32 @ 90%) | 47.0747 ± 0.0193 | 99.7819 ± 0.0028 |
| *GGNN dp* | 98.4356 ± 0.0063 (05/32 @ 90%) | 0.9925 ± 0.0004 | 63.7686 ± 0.0940 |
| *NRI encoder* | 99.8306 ± 0.0024 (14/32 @ 90%) | 43.4380 ± 0.0220 | 99.3851 ± 0.0051 |
| *RL ablation* | 96.6928 ± 0.0131 (02/32 @ 90%) | 94.8530 ± 0.0164 | 97.8541 ± 0.0091 |
| *GFSA Layer (ours)* | 99.6561 ± 0.0030 (25/32 @ 90%) | **98.9355 ± 0.0056** | 99.8973 ± 0.0020 |
| **100 training examples** | | | |
| *RAT nw* | 80.2832 ± 0.0257 (00/32 @ 90%) | 1.1217 ± 0.0021 | 83.4938 ± 0.0284 |
| *GREAT nw* | 78.8755 ± 0.0220 (00/32 @ 90%) | 6.9583 ± 0.0157 | 60.9003 ± 0.0375 |
| *GGNN nw* | 79.3594 ± 0.0350 (00/32 @ 90%) | 28.2760 ± 0.3023 | 5.6617 ± 0.0095 |
| *RAT dp* | 59.5289 ± 0.0174 (00/32 @ 90%) | 28.9121 ± 0.0076 | 62.2680 ± 0.0500 |
| *GREAT dp* | 57.0199 ± 0.0378 (00/32 @ 90%) | 27.1285 ± 0.0161 | 64.4819 ± 0.0339 |
| *GGNN dp* | 44.3653 ± 0.0182 (00/32 @ 90%) | 9.6449 ± 0.0060 | 38.3370 ± 0.0223 |
| *NRI encoder* | 68.6947 ± 0.0390 (00/32 @ 90%) | 26.6422 ± 0.0172 | 73.5216 ± 0.0312 |
| *RL ablation* | 98.4823 ± 0.0087 (06/32 @ 90%) | 97.0341 ± 0.0141 | 99.1689 ± 0.0089 |
| *GFSA Layer (ours)* | **98.8141 ± 0.0069** (25/32 @ 90%) | **97.8198 ± 0.0079** | **99.2172 ± 0.0048** |

| Task | Last Write | | |
|---|---|---|---|
| Example size | 1x | 2x | 0.5x |
| **100,000 training examples** | | | |
| *RAT nw* | 99.8333 ± 0.0021 (22/32 @ 90%) | 94.8665 ± 0.0172 | **99.9741 ± 0.0012** |
| *GREAT nw* | 99.7538 ± 0.0043 (16/32 @ 90%) | 93.2187 ± 0.0181 | 99.9343 ± 0.0022 |
| *GGNN nw* | 98.8240 ± 0.0080 (09/32 @ 90%) | 40.6941 ± 0.0302 | 88.2834 ± 0.0281 |
| *RAT dp* | 99.9815 ± 0.0006 (19/32 @ 90%) | 68.9617 ± 0.0169 | 99.7626 ± 0.0045 |
| *GREAT dp* | **99.9868 ± 0.0007** (18/32 @ 90%) | 68.4564 ± 0.0188 | 99.8809 ± 0.0029 |
| *GGNN dp* | 99.3488 ± 0.0040 (13/32 @ 90%) | 38.3976 ± 0.0772 | 94.5246 ± 0.0576 |
| *NRI encoder* | 99.8710 ± 0.0019 (24/32 @ 90%) | 52.7272 ± 0.0226 | 99.8390 ± 0.0058 |
| *RL ablation* | 98.0828 ± 0.0109 (03/32 @ 90%) | 96.6400 ± 0.0185 | 98.9277 ± 0.0076 |
| *GFSA Layer (ours)* | 99.4653 ± 0.0040 (25/32 @ 90%) | **98.7259 ± 0.0111** | 99.7763 ± 0.0033 |
| **100 training examples** | | | |
| *RAT nw* | 79.2705 ± 0.0212 (00/32 @ 90%) | 8.9069 ± 0.0165 | 83.7914 ± 0.0379 |
| *GREAT nw* | 80.1879 ± 0.0273 (00/32 @ 90%) | 40.2206 ± 0.0386 | 84.5417 ± 0.0312 |
| *GGNN nw* | 91.1302 ± 0.0196 (01/32 @ 90%) | 71.6216 ± 0.0163 | 91.7911 ± 0.0272 |
| *RAT dp* | 75.9944 ± 0.0352 (00/32 @ 90%) | 48.0974 ± 0.0331 | 81.6254 ± 0.0312 |
| *GREAT dp* | 73.6926 ± 0.0391 (00/32 @ 90%) | 46.2676 ± 0.0334 | 80.0267 ± 0.0511 |
| *GGNN dp* | 53.8178 ± 0.0282 (00/32 @ 90%) | 17.8435 ± 0.0101 | 55.0784 ± 0.0481 |
| *NRI encoder* | 65.3841 ± 0.0498 (00/32 @ 90%) | 36.4278 ± 0.0301 | 73.8556 ± 0.0106 |
| *RL ablation* | 98.3220 ± 0.0098 (06/32 @ 90%) | **96.9613 ± 0.0150** | 99.0671 ± 0.0074 |
| *GFSA Layer (ours)* | **98.7144 ± 0.0072** (24/32 @ 90%) | **96.9758 ± 0.0120** | **99.5543 ± 0.0068** |

Figure D.4: Precision-recall curves for a subset of the static analysis experiments that reveals interesting differences in performance: training on 100 examples and evaluating on the same data distribution, and training on 100,000 examples but evaluating on examples of twice the size. Crosshatches indicate candidate thresholds that were evaluated at test time. Best viewed in color.

### D.3 Variable Misuse

#### D.3.1 Dataset

We use the dataset released by Hellendoorn et al. [19], which is derived from a redistributable subset of the ETH 150k Python dataset [33].[7] For each top-level function and class definition extracted from the original dataset, this derived dataset includes up to three modified copies introducing synthetic variable misuse errors, along with an equal number of unmodified copies. For our experiments, we do additional preprocessing to support the GFSA layer: we encode the examples as graphs, and throw out examples with more than 256 nodes or 512 node-observation tuples, which leaves us with 84.5% of the dataset from Hellendoorn et al. [19].

#### D.3.2 Model Architectures

As in the edge classification task, we convert the AST nodes into graph nodes, using the same helper nodes and connectivity structure described in appendix B.1. For this task, when an AST node has multiple children, we add extra edges specifying the index of each child; this is used only by the attention model, not by the GFSA layer. In addition to node features based on the AST node type, we include features based on a bag-of-subtokens representation of each AST node. We use a 10,000-token subword encoder implemented in the `Tensor2Tensor` library by Vaswani et al. [41], pretrain it on GitHub Python code, and use it to tokenize the syntax for each AST node. We then compute node features by summing over the embedding vectors of all subtokens that appear in each node. The learned embedding vectors are of dimension 128, which we project out to 256 before using as node features.

To ensure that we can compare results across different edge types in a fair way, we fix the sizes of the base models. For the RAT and GREAT model families, we use a hidden dimension of 256 and 8 attention heads with a per-head query and value dimension of 32. For the GGNN model family, we use a hidden dimension of 256 and a message dimension of 128. For all models, we use positional embeddings for node features, and edge types embedded as 64-dimensional vectors. We embed all edge types separately in the forward and reverse directions, including both the base AST edges as well as any edges added by learned edge layers; for learned edges we compute new edge features by weighting each embedding vector by the associated edge weight. For the "@ start" edge types, the edges are all embedded at the same time, and for the "@ middle" edge types, we modify the edge features after adding the new edges and use the modified edge features for all following model layers. We compute our final outputs by performing a learned dot-product operation on our final node embeddings $Y$ and then taking a softmax transformation to obtain a distribution over node pairs:

$$ Y = f_\theta(X_{\text{node}}, X_{\text{edge}}), \qquad Z = \text{softmax}\left(\{\boldsymbol{y}_n^T W \boldsymbol{y}_{n'}\}_{n,n' \in N}\right). $$

As described in Vasic et al. [39], we compute a mask that indicates the location of all local variables that could be either bug locations or repair targets (along with the sentinel no-bug location). We then set the entries of $Z_{n,n'}$ to zero for the locations not contained in the mask, and renormalize so that it sums to 1 across node pairs. Note that there is always exactly one correct bug location $n$ but there could be more than one acceptable repair location $n'$; we thus sum over all correct repair locations to compute the total probability assigned to correct bug-repair pairs, and then use the standard cross-entropy loss.

For the GFSA edges, we use an initialization temperature of $\beta = 0.2$, and fix $|Z| = 4$. We use a single finite-state automaton policy to generate two edge types by computing the trajectories when $z_0 = 0$ as well as when $z_0 = 1$. We set $T_{\max} = 128$.

For the NRI head edges, we use a 3-layer MLP (with hidden sizes [32, 32] and output size 2), and take a logistic sigmoid of the outputs, interpreting it as a weighted adjacency matrix for two edge types.

For the uniform random walk edges, we learn a single halting probability $p_{\text{halt}} = \sigma(\theta_{\text{halt}})$ along with adjustment parameters $a, b \in \mathbb{R}$ as defined in section 3.3. The output adjacency matrix is defined

Redistributable subset: `https://github.com/google-research-datasets/eth_py150_open`
With synthetic errors (as released by Hellendoorn et al. [19]):
`https://github.com/google-research-datasets/great`

similarly to the GFSA model, but with all of the policy parameters fixed to move to a random neighbor with probability $1 - p_{\text{halt}}$ and take the ADDEDGEANDSTOP action with probability $p_{\text{halt}}$. For this model, we only add a single edge type.

The RL agent uses the same parameterization as the GFSA layer, but samples a single trajectory for each start node and uses it to add a single edge (or no edge) from each start node. The downstream cross-entropy loss for the classification model is used as the reward for all of these trajectories. Since simply computing this reward requires a full downstream model forward pass, we run only one rollout per example with a learned scalar reward baseline $\hat{R}$. We add an additional loss term $\alpha(R - \hat{R})^2$ so that this learned baseline approximates the expected reward, and scale the REINFORCE gradient term by a hyperparameter $\beta$.

The "Hand-engineered edges" baseline uses the base AST edges and adds the following edge types from Allamanis et al. [1] and Hellendoorn et al. [19]: NextControlFlow, ComputedFrom, FormalArgName, LastLexicalUse, LastRead, LastWrite, NextToken (connecting syntactically adjacent nodes), Calls (connecting function calls to their definitions), and ReturnsTo (connecting return statements to the function they return from).

### D.3.3  Training and Detailed Results

For all of our models, we train using the Adam optimizer for 400,000 iterations; this is enough time for all models to converge to their final accuracy. We use a batch size of 64 examples, grouping examples of similar size to avoid excessive padding.

For each model, we randomly sample 32 hyperparameter settings for the learning rate (log-uniform in $[10^{-5}, 10^{-2}]$) and gradient clipping threshold (log-uniform in $[1, 10^4]$). For the GFSA models, we also tune $\varepsilon_{\text{bt-stop}}$ (log-uniform in $[0.001, 0.1]$). For the RL ablation, we tune the weight of the relative weights of different gradient terms: $\alpha$ is chosen log-uniformly in $[0.00001, 0.1]$ and $\beta$ is chosen in $[0.001, 2.0]$. Over the course of training, we take a subset of approximately 7000 validation examples and compute the top-1 accuracy of each model on this subset. We then choose the hyperparameter settings and early-stopping point with the highest accuracy.

We evaluate the selected models on the size-filtered test set, containing 818,560 examples. For each example and each model, we determine the predicted classification by determining whether 50% or more probability is assigned to the no-bug location. For incorrect examples, we then find the pair of predicted bug location and repair identifier with the highest probability (summing over all locations for each candidate repair identifier), and check whether the bug location and replacement identifier are correct. Note that if the model assigns >50% probability to the no-bug location, but still ranks the true bug and replacement highest with the remaining probability mass, we count that as an incorrect classification but a correct localization and repair.

To compute standard error estimates, we assume that predictions are independent across different functions, but may be correlated across modified copies of the same function; we thus estimate standard error by using the analytic variance for a binomial distribution, adjusted by a factor of 3 (for buggy or non-buggy examples analyzed separately) or 6 (for averages across all examples) to account for the multiple copies of each function in the dataset. Table D.2 contains higher-precision results for the variable misuse tasks, along with standard error estimates, a breakdown of marginal localization and repair scores (examples where the model gets one of the locations correct but possibly the other incorrect), and an overall accuracy score capturing classification, localization, and repair.

Table D.2: Full-precision results on variable misuse task, with additional breakdown of accuracy for buggy examples. Expressed as accuracy (in %) ± standard error.

| Example type: | All | | No bug | With bug |
|---|---|---|---|---|
| | Classification | Class & Loc & Rep | Classification | Classification |
| **RAT** | | | | |
| *Base AST graph only* | 92.540 ± 0.071 | 88.225 ± 0.087 | 92.051 ± 0.073 | 93.030 ± 0.069 |
| *Base AST graph, +2 layers* | 92.245 ± 0.072 | 87.846 ± 0.088 | 92.455 ± 0.072 | 92.035 ± 0.073 |
| *Hand-engineered edges* | 92.704 ± 0.070 | 88.496 ± 0.086 | 92.932 ± 0.069 | 92.477 ± 0.071 |
| *NRI head @ start* | 92.880 ± 0.070 | 88.710 ± 0.085 | 92.551 ± 0.071 | 93.208 ± 0.068 |
| *NRI head @ middle* | 92.572 ± 0.071 | 88.423 ± 0.086 | 92.834 ± 0.070 | 92.310 ± 0.072 |
| *Random walk @ start* | 92.997 ± 0.069 | 88.907 ± 0.084 | **93.224 ± 0.068** | 92.770 ± 0.070 |
| *RL ablation @ middle* | 92.036 ± 0.073 | 87.278 ± 0.090 | 90.361 ± 0.080 | 93.711 ± 0.066 |
| *GFSA layer (ours) @ start* | **93.328 ± 0.068** | **89.472 ± 0.083** | **93.101 ± 0.069** | 93.555 ± 0.066 |
| *GFSA layer (ours) @ middle* | **93.456 ± 0.067** | **89.627 ± 0.082** | 92.662 ± 0.071 | **94.250 ± 0.063** |
| **GREAT** | | | | |
| *Base AST graph only* | 91.662 ± 0.075 | 86.906 ± 0.091 | 90.849 ± 0.078 | 92.475 ± 0.071 |
| *Base AST graph, +2 layers* | 92.307 ± 0.072 | 87.902 ± 0.087 | **92.711 ± 0.070** | 91.903 ± 0.074 |
| *Hand-engineered edges* | 92.287 ± 0.072 | 87.646 ± 0.088 | 92.577 ± 0.071 | 91.996 ± 0.073 |
| *NRI head @ start* | 92.061 ± 0.073 | 87.447 ± 0.089 | 91.112 ± 0.077 | 93.009 ± 0.069 |
| *NRI head @ middle* | 92.074 ± 0.073 | 87.552 ± 0.088 | **92.800 ± 0.070** | 91.347 ± 0.076 |
| *Random walk @ start* | 92.644 ± 0.071 | 88.283 ± 0.087 | 91.872 ± 0.074 | **93.417 ± 0.067** |
| *RL ablation @ middle* | 91.707 ± 0.075 | 86.939 ± 0.091 | 89.951 ± 0.081 | **93.464 ± 0.067** |
| *GFSA layer (ours) @ start* | **92.963 ± 0.069** | **88.825 ± 0.085** | **92.872 ± 0.070** | 93.055 ± 0.069 |
| *GFSA layer (ours) @ middle* | **93.019 ± 0.069** | **88.806 ± 0.085** | 92.427 ± 0.072 | **93.612 ± 0.066** |
| **GGNN** | | | | |
| *Base AST graph only* | 89.704 ± 0.082 | 83.521 ± 0.098 | **91.257 ± 0.076** | 88.152 ± 0.087 |
| *Base AST graph, +2 layers* | 90.359 ± 0.080 | 84.383 ± 0.098 | 88.795 ± 0.085 | **91.922 ± 0.074** |
| *Hand-engineered edges* | **90.874 ± 0.078** | **84.776 ± 0.096** | 90.187 ± 0.081 | 91.560 ± 0.075 |
| *NRI head @ start* | 90.433 ± 0.080 | 84.473 ± 0.096 | **91.486 ± 0.076** | 89.380 ± 0.083 |
| *NRI head @ middle* | 90.243 ± 0.080 | 84.412 ± 0.098 | 88.289 ± 0.087 | **92.198 ± 0.073** |
| *Random walk @ start* | 90.315 ± 0.080 | 84.519 ± 0.096 | **91.351 ± 0.076** | 89.278 ± 0.084 |
| *RL ablation @ middle* | 90.540 ± 0.079 | **84.959 ± 0.096** | 90.437 ± 0.080 | 90.643 ± 0.079 |
| *GFSA layer (ours) @ start* | **90.939 ± 0.078** | **85.012 ± 0.096** | 90.083 ± 0.081 | 91.796 ± 0.074 |
| *GFSA layer (ours) @ middle* | 90.394 ± 0.080 | **84.723 ± 0.096** | 90.983 ± 0.078 | 89.805 ± 0.082 |

| Example type: | With bug | | | |
|---|---|---|---|---|
| | Localization | Repair | Loc & Repair | Class & Loc & Rep |
| **RAT** | | | | |
| *Base AST graph only* | 92.936 ± 0.069 | 91.892 ± 0.074 | 88.300 ± 0.087 | 84.399 ± 0.098 |
| *Base AST graph, +2 layers* | 92.638 ± 0.071 | 91.541 ± 0.075 | 87.764 ± 0.089 | 83.238 ± 0.101 |
| *Hand-engineered edges* | 93.100 ± 0.069 | 92.007 ± 0.073 | 88.388 ± 0.087 | 84.060 ± 0.099 |
| *NRI head @ start* | 93.180 ± 0.068 | 92.304 ± 0.072 | 88.731 ± 0.086 | 84.869 ± 0.097 |
| *NRI head @ middle* | 93.013 ± 0.069 | 92.176 ± 0.073 | 88.619 ± 0.086 | 84.011 ± 0.099 |
| *Random walk @ start* | 93.227 ± 0.068 | 92.282 ± 0.072 | 88.726 ± 0.086 | 84.590 ± 0.098 |
| *RL ablation @ middle* | 92.553 ± 0.071 | 91.606 ± 0.075 | 87.730 ± 0.089 | 84.195 ± 0.099 |
| *GFSA layer (ours) @ start* | 93.820 ± 0.065 | 92.834 ± 0.070 | 89.577 ± 0.083 | 85.843 ± 0.094 |
| *GFSA layer (ours) @ middle* | **94.058 ± 0.064** | **93.083 ± 0.069** | 89.932 ± 0.081 | **86.593 ± 0.092** |
| **GREAT** | | | | |
| *Base AST graph only* | 92.030 ± 0.073 | 91.156 ± 0.077 | 87.179 ± 0.091 | 82.964 ± 0.102 |
| *Base AST graph, +2 layers* | 92.585 ± 0.071 | 91.477 ± 0.076 | 87.698 ± 0.089 | 83.093 ± 0.101 |
| *Hand-engineered edges* | 92.174 ± 0.073 | 91.287 ± 0.076 | 87.168 ± 0.091 | 82.715 ± 0.102 |
| *NRI head @ start* | 92.446 ± 0.072 | 91.520 ± 0.075 | 87.628 ± 0.089 | 83.781 ± 0.100 |
| *NRI head @ middle* | 92.213 ± 0.073 | 91.166 ± 0.077 | 87.258 ± 0.090 | 82.303 ± 0.103 |
| *Random walk @ start* | 92.849 ± 0.070 | 91.949 ± 0.074 | 88.272 ± 0.087 | 84.694 ± 0.097 |
| *RL ablation @ middle* | 92.383 ± 0.072 | 91.295 ± 0.076 | 87.486 ± 0.090 | 83.927 ± 0.099 |
| *GFSA layer (ours) @ start* | **93.466 ± 0.067** | 92.279 ± 0.072 | **88.845 ± 0.085** | 84.779 ± 0.097 |
| *GFSA layer (ours) @ middle* | 93.266 ± 0.068 | 92.394 ± 0.072 | **88.863 ± 0.085** | 85.186 ± 0.096 |
| **GGNN** | | | | |
| *Base AST graph only* | 89.243 ± 0.084 | 87.703 ± 0.089 | 81.633 ± 0.105 | 75.785 ± 0.116 |
| *Base AST graph, +2 layers* | **90.633 ± 0.079** | 88.948 ± 0.085 | 83.969 ± 0.099 | 79.972 ± 0.108 |
| *Hand-engineered edges* | **90.681 ± 0.079** | 88.770 ± 0.085 | 83.524 ± 0.100 | 79.365 ± 0.110 |
| *NRI head @ start* | 89.915 ± 0.082 | 88.151 ± 0.087 | 82.731 ± 0.102 | 77.460 ± 0.113 |
| *NRI head @ middle* | 90.352 ± 0.080 | **89.613 ± 0.083** | **84.443 ± 0.098** | **80.535 ± 0.107** |
| *Random walk @ start* | 89.729 ± 0.082 | 88.611 ± 0.086 | 82.956 ± 0.102 | 77.688 ± 0.113 |
| *RL ablation @ middle* | 90.560 ± 0.079 | 89.269 ± 0.084 | **84.301 ± 0.098** | 79.480 ± 0.109 |
| *GFSA layer (ours) @ start* | **90.939 ± 0.078** | 88.960 ± 0.085 | 83.909 ± 0.099 | 79.942 ± 0.108 |
| *GFSA layer (ours) @ middle* | 90.217 ± 0.080 | 88.886 ± 0.085 | 83.633 ± 0.100 | 78.463 ± 0.111 |

## Footnotes

[5] `https://github.com/serge-sans-paille/gast/releases/tag/0.3.3`

[6]`https://cloud.google.com/tpu/`

[7] Original Python corpus (from Raychev et al. [33]): `https://www.sri.inf.ethz.ch/py150`