[Reviews · NeurIPS 2020]

Review 1

Summary and Contributions: * A differentiable method based on finite automata is presented to find new (useful) edges in a graph. * Experiments grid-world navigation and a program analysis task show that the method succeeds at learning relevant relationships that help downstream models.

Strengths: * While composed of well-understood components, the proposed method is new in the context of learning from graphs. * The method is sound and seems to provide (minor) gains on the well-established variable misuse task, competing with strong baselines. * Methods to automatically discover important relationships in input data are of high relevance to the NeurIPS community.

Weaknesses: The gains in the experiments are minor, and the most substantial part of the experiments is limited to the program analysis setting, in which the target relationships are well understood. Further experiments on other domains (e.g., on sequences encoding proteins (cf. https://blog.einstein.ai/provis/) or graphs of theorems)) would be useful to better understand the generality of the proposed approach. This would have the potential to turn this from an interesting, specialized paper into a seminal paper.

Correctness: Claims made in the paper seem correct, and are supported by proofs in the appendix.

Clarity: The paper is relatively well-written, though dense. Many details can only be found in the somewhat sprawling appendix. Minor nits: * L60-61: "one can construct a regular language L such that an L-path from n_1 to n_2 corresponds to the location n_2 of possible previous assignments to the variable at n_1" sounds weird (singular/plural disagreement), and should probably read "one can construct a regular language L such that for a variable at location n_1 and a previous assignment to that variable at location n_2, there exists an L-path from n_1 to n_2"

Relation to Prior Work: The relationship to prior work is clearly articulated, though I am missing some discussion of the relationship of the proposed method to the implicit learning of relationships in transformers. Specifically, it is well-known that attention weights can be viewed as a weighted adjacency matrix. While I could conjecture some differences to the method proposed here (e.g., transformers only do one-step reasoning [but what about deep transformer models / universal transformer?]), the paper would be improved by discussing this as well.

Reproducibility: Yes

Additional Feedback:


Review 2

Summary and Contributions: The paper introduces a differentiable approach to automatically compute semantically meaningful long-distance edges over static graphs. It can be used during end-to-end learning of graph-based (GNN or Transformer driven) tasks. The work observes that many such hand-crafted edges can be represented as regular languages over graph paths, and thus can be encoded by a POMDP following agent. It then proposes an efficient procedure for training such an agent as a differentiable module in a larger graph network, using its halting distribution as a new weighted edge type. The work is evaluated against several baselines on ML4Code tasks (either learning supervised semantic edges or learning an end-to-end task), and noticeably outperforms them using less training data.

Strengths: + An insightful novel formulation for learning semantic graph edges end-to-end, without hand-engineering them (e.g. as data flow). + Theoretical analysis that proves validity of the approach for any regular language of the target semantic edge – in particular, for data flow edges of program representations. + Comprehensive and excellent evaluation on three distinct families of tasks against strong baselines. + Clear benefit of the inductive bias with less training data. + The paper is well written and accessible.

Weaknesses: - Some missing or unacknowledged related work.

Correctness: The theoretical findings and empirical methodology seem sound. I did not check the Appendix proofs in detail.

Clarity: I'm well familiar with the ML4Code literature, less so with the RL option literature. The paper is excellently presented for such an audience, introducing parts of its approach step by step, with examples, and appropriate references. It's a pleasure to read. All the relevant technical details are described in the Appendix in elaborate detail. For an audience less familiar with ML4Code, I suggest introducing a new figure in the Introduction that presents the code snippet from Fig 1b with static (handcrafted) data-flow edges, contrasted side-by-side with the edges learned by GFSA (similar figure to existing Fig 1b). This will also help define the concept of data flow (L60-67).

Relation to Prior Work: The prior work is adequately discussed, with one significant exception and a couple minor connections. Significant: What the authors call "RelAtt" was introduced by Wang et al. [1] concurrently with the GREAT model. They call it RAT (Relation-Aware Transformer) and apply on the task on encoding database schema graphs rather than encoding program AST graphs. Minor: - The authors might want to discuss a possible connection of GFSA to Neural State Machines [2], although the connection is weak. - GFSA learns a POMDP as an approximation of the latent FSA representing the desired edge relation. One could also imagine constructing such an FSA explicitly [3] and either (a) using it as a form of supervision to GFSA, or (b) relaxing the FSA into differentiable form to facilitate end-to-end learning akin to GFSA. This requires knowing the desired edge relation, thus might not be possible for an end-to-end task like VarMisuse, though. [1] Wang, B., Shin, R., Liu, X., Polozov, O. and Richardson, M., 2020. RAT-SQL: Relation-aware schema encoding and linking for text-to-SQL parsers. In Proceedings of the 58th Annual Meeting of the Association for Computational Linguistics (ACL). [2] Hudson, D. and Manning, C.D., 2019. Learning by abstraction: The neural state machine. In Advances in Neural Information Processing Systems (pp. 5903-5916). [3] Weiss, G., Goldberg, Y. and Yahav, E., 2018, July. Extracting automata from recurrent neural networks using queries and counterexamples. In International Conference on Machine Learning (pp. 5247-5256).

Reproducibility: Yes

Additional Feedback: This is great work and should definitely be accepted. A few thoughts and comments on the follow-up: * The fact that observations depend on the initial node (L114) introduces a limited form of "variable capturing" in the regular language that the POMDP tries to approximate. It is still regular, but this can be broadened to depend on the whole agent's history and thus represent more complex languages. The POMDP would no longer be theoretically guaranteed to represent such a language, but it might still learn a useful one. * The VarMisuse experiments show that GFSA-enriched networks seem to use the new weighted edges as extra representations, inducing some form of long-distance structure in the input graph (akin to Transformer attention structure). I wonder what would happen if one used GFSA for self-supervised program learning like CodeBERT [4] rather than supervised program analysis tasks. Training on a large corpus of _natural_ (rather than generated) code with a self-supervised objective might induce new semantic relations that might shed new light on which long-distance dependencies are actually important to capture an informative representation of the code. * Handcrafted data flow edges are sharp and discrete. GFSA-learned probability matrices are not necessarily so. Have you considered any regularization on sharpness of the probabilities? [4] Feng, Z., Guo, D., Tang, D., Duan, N., Feng, X., Gong, M., Shou, L., Qin, B., Liu, T., Jiang, D. and Zhou, M., 2020. CodeBERT: A pre-trained model for programming and natural languages. arXiv preprint arXiv:2002.08155.


Review 3

Summary and Contributions: This paper addresses the problem of augmenting graph data with additional edges corresponding to higher-level abstract relations. For example, an abstract syntax tree (AST) might be augmented with edges that identify the last statement to read a variable. The authors propose an end-to-end differentiable method to learn policies for finding these additional edges by casting the problem as a POMDP for a finite state machine (whose corresponding regular expressions characterize the paths of interest) and deriving a differentiable policy for navigating it from a source state to an absorbing state. This policy can then be used to more quickly solve a downstream task that depends on the derived edges.

Strengths: -The paper is clearly written and definitely of relevance to the NeurIPS community. -The authors characterize the problem nicely in terms of existing RL theory (POMDPs) and establish results which map learning in this environment to paths in a finite state automaton. They also connect their analysis to finding successor states in an RL problem. -The method is evaluated on several tasks including a GridWorld problem and program analysis and the results seem moderately better than baselines (especially for less training data)

Weaknesses: -The results are somewhat mixed and the domains are relatively simple -There is another body of work which I think is related to this but not mentioned in the automatic knowledge-base completion literature. In particular I am curious how this method compares to something like: Go For a Walk: https://arxiv.org/abs/1711.05851 (other than the fact that GFW uses RL explicitly). Could you not compare to existing RL approaches?

Correctness: As far as I can tell everything looks correct.

Clarity: The paper is well written and reasonable clear. Because it focuses on program analysis, there may be a bit of a learning curve for readers not familiar with that literature.

Relation to Prior Work: The prior work section seems complete, except for coverage of RL-based path discovery work mentioned above.

Reproducibility: Yes

Additional Feedback: Thank you for running the GFAW comparison. I have raised my score.


Review 4

Summary and Contributions: In this paper, the authors study the problem of how to learn to add edges into input graphs for better performance in downstream tasks. The main contribution are summarized as follows. C1. The authors formulate the problem of learning to add edges into input graphs, and highlight the potential applications in software engineering scenario. C2. GFSA is proposed to enable edge addition learning from downstream tasks. C3. Empirical results on program analysis and bug detection tasks suggest the effectiveness of the proposed method.

Strengths: S1. The authors investigate an interesting aspect of graph learning: how to enable edge addition learning with context understanding. S2. The authors also suggest a unique angle to address the problem of context understanding. In particular, context understanding is formulated as a process of state transition triggered by actions in a finite-state automaton, and casted into a learning problem. S3. The effectiveness of the proposed method is supported by multiple concrete tasks in software engineering and program analysis, indicating its real-world impact.

Weaknesses: W1. The unique value of the propose GFSA is unclear. As discussed in the paper, the problem setting behind GFSA is friendly to reinforcement learning techniques. There has been some work that aims to learn better graph structure for downstream tasks by reinforcement learning [1]. The key questions is why GFSA is preferred over existing reinforcement learning techniques. Without the clarification, the unique value in GFSA remains vague. W2. The action "backtrack" could be problematic. While this action enables the model to handle the cases where no edge addition is needed or discovered, it could also make the model run into endless loops. How does the final reward guide the model to minimize the chance of triggering "backtrack"? W3. The proposed method may suffer low learning efficiency. As reward comes from downstream tasks, supervision on early actions could be fairly weak. How is this issue addressed in this work? W4. It is difficult to see how GFSA could support multiple edge addition. Reference [1] Wang, L. et al. Learning Robust Representations with Graph Denoising Policy Network.

Correctness: Yes

Clarity: Yes

Relation to Prior Work: Yes

Reproducibility: Yes

Additional Feedback: Q1. Does GFSA only support one edge addition? Q2. In practice, for two nodes connected by an added edge, what is their distance? If they are far away, it means many actions are needed to reach that edge addition decision, and the issue discussed in W3 could be severe. If they are local, why not casting the context understanding as a link prediction problem and solving it by message passing models (e.g., GNN variants)? ======After rebuttal====== The authors shared reasonable response to my questions. I have raised the score.

[Author Response · NeurIPS 2020]

We would like to thank the reviewers for their comments, and in particular appreciate the insightful suggestions for
future work by R1 and R2. We start by focusing on addressing significant misunderstandings and unfounded assertions,
then describe some additional experiments, and finally discuss some of the other comments and questions.

**Misunderstandings:**

• **R4:** *"The proposed method may suffer low learning efficiency. As reward comes from downstream tasks, supervision
on early actions could be fairly weak."* This is unfounded, and it appears the reviewer has misunderstood critical
details. A central idea in GFSA is to analytically solve for the marginal distribution over halting behavior, and to
directly differentiate through the solution with implicit differentiation (Sec 3.2). There is no reason why supervision
on early actions would be weak. Table 1 shows that GFSA has excellent sample efficiency, and Fig 1 shows an
example of many-step behavior. The LASTREAD policy shown there takes an average of 35 actions before accepting.

• **R4:** *"It is difficult to see how GFSA could support multiple edge addition."* It's unclear if the reviewer means multiple
edges per graph (GFSA does this), multiple edges per start node (GFSA does this, but we will add a sentence to
emphasize this), or multiple edge types (GFSA does this using automaton states).

• **R4:** *"The action 'backtrack' could be problematic. While this action enables the model to handle the cases where no
edge addition is needed or discovered, it could also make the model run into endless loops."* It is not problematic. As
long as the policy places positive probability on STOP and/or ADDEDGEANDSTOP, the Markov chain will terminate
with probability 1. In practice, when solving for marginals, we treat backtracking as termination and then correct for
backtracking as a post-processing step (See L569-576 in Appendix).

• **R4:** *"How does the final reward guide the model to minimize the chance of triggering 'backtrack'?"* We emphasize
that the GFSA layer is formulated in terms of a POMDP, but we are not using RL and have no reward function. The
GFSA layer is just a deterministic, differentiable layer used within a supervised learning architecture. The end-to-end
loss encourages useful ADDEDGEANDSTOP actions, so no explicit "don't backtrack" signal is needed.

**Additional experiments and differences from RL approaches:** At a fundamental level, the GFSA layer and RL
approaches have different types of output. The GFSA layer produces a full distribution (represented as a continuous-
valued vector) that can be transformed nonlinearly (e.g. $f(\mathbb{E}[\tau])$ where $f$ is the downstream model and loss and $\tau$ are
edge additions from trajectories). In contrast, RL approaches produce stochastic discrete samples. As such, it is not
possible to "drop in" a standard RL approach instead of GFSA; one must first reformulate the model, task, and training
loop in terms of expected reward and discrete latent variables ($\mathbb{E}[f(\tau)]$). Nevertheless, we ran an experiment inspired by
the "Go For a Walk" paper suggested by R3, training a standard RL agent with the same parameterization as GFSA on
modified versions of our tasks. For the program analysis tasks, we replace the cross-entropy loss with a reward of +1 for
adding a correct edge (or correctly not adding any) and 0 otherwise. We train using REINFORCE with 20 rollouts per
start node and a leave-one-out control variate. It suffers from high variance, and the best version underperforms GFSA.
For 100k examples, 1x size: 94.2% v.s. 100%, 96.7% v.s. 99.6%, 98.1% v.s. 99.5%. Because edges are added by single
trajectories rather than marginals over trajectories (as in GFSA), these agents are unable to learn to add multiple edges
per start node. For the variable misuse task, we use the final classification log-likelihood as a reward. Simply computing
this reward requires a full downstream model forward pass, so we run only one rollout per example with a learned scalar
reward baseline. The best model with these RL-based edges performs similarly to the model on the base AST graph
alone, and does not learn to add useful edges. We will describe these experiments in more detail in the next revision.

**Prior work:** We would like to thank R2, R3, and R4 for their additional citations, and especially R2 for the Relation-
Aware Transformer paper, which indeed appears to be equivalent to the "RelAtt" model. We will include these in the
next revision, along with a discussion of GFSA v.s. transformer attention as suggested by R1.

**Experimental gains:** For the variable misuse task, although the gains of our method are only a few percent, even the
hand-engineered edges from previous work lead to only a few percent improvement over the base graph after a thorough
hyperparameter search. We have thus shown that an end-to-end-trained model can outperform hand-engineered edges
for these tasks, and verified that these differences are statistically significant on our test set.

**Future work:** Thanks R1 and R2 for interesting discussions and suggestions about how to take this work forward.
We agree with R1 that applying GFSA ideas to other domains is an exciting next step. Similarly, R2's suggestion
to consider pairing GFSA with code-based self-supervised pre-training is a really interesting suggestion. We hadn't
thought of framing the start point-specific observations as a form of variable capture, but that's a nice way of thinking
about it. Conditioning on the initial node is straightforward because we can still marginalize out the agent's history
while solving for the distribution. Modifying the language based on agent history would make this harder, but might
be tractable if we could enumerate the possible "captures" and solve them jointly. A related idea we've considered is
adding a stack of states, corresponding to context-free grammars; we hope to explore this and similar ideas in the future.
We also hope to explore methods for encouraging sparsity; we ran some experiments with entropy regularization after
the paper submission but unfortunately this decreased performance on the downstream task.

[Meta-Review · NeurIPS 2020]

In graph nets, edges can represent two kinds of relations: ones that follow immediately from the structure of the graph, and ones that are abstract/implicit. The paper proposes to learn the latter. More precisely, it considers relations defined as paths in the base graph accepted by a finite-state automaton, poses the problem of learning these relations as a POMDP problem, and solves a relaxed version of this problem using gradient descent. Overall, the paper was well-received. Pros: + Fresh idea + Clean formulation + Experiments show clear gains in the domains considered + The paper is well-written Cons: - Some missing related work - Somewhat narrow application domain The reviewers appreciated the clarifications provided in the author response, in particular the RL experiment for the "Go for a Walk" domain. Please integrate your responses in the rebuttal with the main paper. And naturally, consult the reviews for more detailed feedback.